# PLATE: Plasticity-Tunable Efficient Adapters for Geometry-Aware Continual Learning

**Romain Cosentino** [1]

## Abstract

We develop a continual learning method for pre-trained models that *requires no access to old-task data*, addressing a practical barrier in foundation model adaptation where pretraining distributions are often unavailable. Our key observation is that pretrained networks exhibit substantial *geometric redundancy*, and that this redundancy can be exploited in two complementary ways. First, redundant neurons provide a proxy for dominant pretraining-era feature directions, enabling the construction of approximately protected update subspaces directly from pretrained weights. Second, redundancy offers a natural bias for *where* to place plasticity: by restricting updates to a subset of redundant neurons and constraining the remaining degrees of freedom, we obtain update families with reduced functional drift on the old-data distribution and improved worst-case retention guarantees. These insights lead to PLATE (**Pla**sticity-**T**unable **E**fficient Adapters), a continual learning method requiring no past-task data that provides explicit control over the plasticity-retention trade-off. PLATE parameterizes each layer with a structured low-rank update $\Delta W = BAQ^\top$, where $B$ and $Q$ are computed once from pretrained weights and kept frozen, and only $A$ is trained on the new task. Code is available at https://github.com/SalesforceAIResearch/PLATE.

## 1. Introduction

Deep neural networks trained sequentially on multiple tasks tend to forget what they have learned before: performance on old tasks degrades as the model is updated on new data (McCloskey & Cohen, 1989; Ratcliff, 1990; French, 1999; Goodfellow et al., 2013; Parisi et al., 2018). In modern practice, a large backbone is first pretrained on a massive, opaque distribution, and is then adapted to downstream tasks via instruction tuning, domain adaptation, or reinforcement learning (Ettinger et al., 2025). Fully fine-tuning all parameters on each new task is often too expensive and can severely erode core capabilities such as factual knowledge, reasoning, or instruction following.

Parameter-efficient fine-tuning (PEFT) has become the de facto solution to the cost side of this problem: instead of updating all parameters, one modifies only a small subset or a low-dimensional subspace of them using so called adapters (Houlsby et al., 2019; Li & Liang, 2021; Lester et al., 2021; Hu et al., 2021; He et al., 2021). These methods achieve strong performance on new tasks while training and storing only a small fraction of the backbone weights. However, PEFT alone does not eliminate catastrophic forgetting nor ensure the preservation of prior capabilities: even when only adapter parameters are trained, fine-tuning still erodes pretraining-era behavior and generalization (Qiao & Mahdavi, 2024; Zhao et al., 2024; Kalajdzievski, 2024). Recent work has therefore begun to study continual learning specifically under PEFT. For instance, by using small auxiliary "context" sets to shape knowledge-preserving adapter subspaces (Yang et al., 2024b) or forcing orthogonality between successive tasks (Wang et al., 2023). In contrast, our approach targets the fully data-free setting with respect to the old-task distribution: both the protected input subspace and the subset of trainable redundant neurons are inferred directly from frozen pretrained weights. To the best of our knowledge, PLATE is among the first continual-learning methods to derive *both* ingredients in a fully weight-only manner.

A natural way to mitigate forgetting is to constrain new-task updates so that they are "invisible" to the old task. Existing continual-learning methods implement this idea in different ways. Regularization-based approaches such as Elastic Weight Consolidation (Kirkpatrick et al., 2016), Synaptic Intelligence (Zenke et al., 2017), and Memory Aware Synapses (Aljundi et al., 2017) penalize movement along directions that were important for previous tasks. Replay and constrained-gradient methods (Lopez-Paz & Ranzato, 2017; Chaudhry et al., 2018), use stored examples from old

[1]Salesforce AI Research. Correspondence to: Romain Cosentino <rom.cosentino@gmail.com>.

*Proceedings of the 43rd International Conference on Machine Learning*, Seoul, South Korea. PMLR 306, 2026. Copyright 2026 by the author(s).

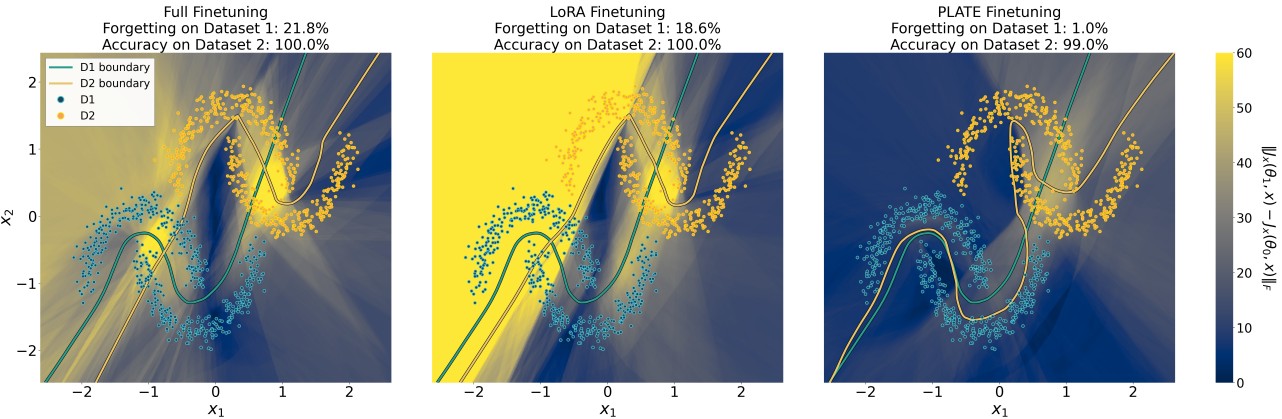

*Figure 1.* **Local-geometry view of forgetting on a continual learning** 2**-dimensional binary classification problem:** Blue points denote the old-task dataset $P_0$ and yellow points the new-task dataset $P_1$; decision boundaries are shown when trained on $P_0$ (blue curve) and after training on $P_1$ (yellow curve). The background heatmap visualizes how training on $P_1$ changes the model's local input-output linearization, $\Delta(x) := \|J_x(\theta_1, x) - J_x(\theta_0, x)\|_F$, where $J_x(\theta, x) := \nabla_x f_\theta(x)$ denotes the input Jacobian; this is distinct from the parameter Jacobian $J_\theta(x) := \nabla_\theta f_\theta(x)$ used throughout Sections 2.2–3.2. *Retention is compromised when the heatmap turns yellow around the blue points* (large drift on $\mathrm{supp}(P_0)$), while *effective learning requires yellow regions around the yellow points* (large drift concentrated near $\mathrm{supp}(P_1)$). *This motivates our goal: parameter-efficient continual updates that localize drift away from the (often unavailable) old distribution while remaining expressive on the new task.* (**Left**) Full fine-tuning induces large changes throughout, including on $\mathrm{supp}(P_0)$, and the old boundary drifts. (**Middle**) LoRA restricts the parameter update but still produces substantial change on $\mathrm{supp}(P_0)$. (**Right**) PLATE updates keep $\Delta(x)$ small on $\mathrm{supp}(P_0)$ while permitting large changes near $P_1$, concentrating plasticity where it is needed and preserving old behavior (see Figure 18 for PLATE hyperparameters sweep).

tasks to project gradients and reduce interference. Explicit orthogonality-based approaches such as Orthogonal Weight Modification and Orthogonal Gradient Descent (Zeng et al., 2018; Farajtabar et al., 2019; Saha et al., 2021) estimate the feature subspace used by old tasks and project new gradients into its orthogonal complement. Architectural methods like Progressive Networks (Rusu et al., 2016) and Dynamically Expandable Networks (Yoon et al., 2017) go further by freezing old parameters and routing new tasks through newly added capacity. Mixture-of-Experts architectures have also been explored for continual learning by routing different tasks/inputs to specialized experts, reducing interference across tasks (Li et al., 2024).

These strategies reduce forgetting to some extent, but they face serious limitations in the regime that motivates contemporary training approaches. First, most methods assume continued access to old-task data (or at least stored examples/gradients/features). In large-scale models, this assumption is often violated: pretraining data are often proprietary, massive, and unavailable. Second, many approaches operate at the level of global gradients over the full model and are not computationally efficient at Large Language Model (LLM) scale. Third, even when old data are available, orthogonality constraints are inherently approximate: whether one estimates protected subspaces from stored data or from curvature surrogates such as Fisher information, feature statistics are noisy, nullspaces are only identified up to finite-sample and numerical error, and projections can only be enforced approximately. Finally, they are usually

not applicable to the LLM scale because of their inherent computational burden.

In this work we ask: *Can we design a continual learning method that reduces forgetting without access to prior-task data, while remaining practical at scale?*

Our approach is motivated by two observations. First, orthogonality-based constraints alone do not eliminate forgetting: if the allowed update family is only approximately orthogonal to old-task features, then there exist directions within it that still induce a non-trivial increase in the old-task loss. Second, pretrained models exhibit substantial redundancy: many neurons implement highly similar features, often visible as strong colinearity among weight vectors, a phenomenon induced by overparameterization, dropout, and the implicit bias of training dynamics (Srivastava et al., 2014; Balestriero et al., 2019; 2021).

This suggests two complementary opportunities for continual learning without past-task data: ($i$) redundancy can serve as a *weight-only proxy* for dominant old-task feature directions, enabling approximate protected subspaces computed directly from frozen weights, ($ii$) redundancy can guide *where to place plasticity*: concentrating updates on redundant neurons biases adaptation toward directions that are less function-changing on the pre-trained distribution.

These insights motivate PLATE (**Pla**sticity-**T**unable **E**fficient Adapters), a PEFT method that implements exactly this recipe. At each layer, before training, PLATE ($i$) identifies a set of trainable redundant neurons from frozen

weights, $(ii)$ builds a low-energy input subspace from the pre-trained weights, both contributing to the protection of the pre-training distribution of the model. We parameterize the model updates as a low-rank adapter: concretely, each adapted layer uses updates of the form $B^{(\ell)} A^{(\ell)} Q^{(\ell)\top}$, where $B^{(\ell)}$ (frozen) selects redundant output neurons, $Q^{(\ell)}$ (frozen) spans a weight-based approximate pre-training distribution nullspace, and $A^{(\ell)}$ are trainable parameters. This yields a parameter efficient geometry-aware adapter that requires no access to pretraining data, exposes an explicit plasticity trade-off via the estimated (resp. selected) input (resp. output) dimensions of the $A$ matrix, and inherits the computational benefits of PEFT methods.

The paper is organized as follows: $(i)$ We show that any approximately protected update family admits a nonzero worst-case forgetting floor and provide a weight-only construction of approximate protected directions from pretrained redundancy (Sections 2). $(ii)$ We relate worst-case forgetting over an update family to old-task restricted curvature, and upper bound this curvature by first-order functional drift (Sections 3). $(iii)$ We propose PLATE, a data-free continual PEFT adapter $\Delta W = BAQ^\top$ that concentrates plasticity on redundant channels and restricts updates to a low-energy input subspace computed once from frozen weights (Section 4). $(iv)$ Across controlled continual-learning benchmarks and LLM specialization, PLATE improves retention over LoRA (Hu et al., 2021) at similar trainable budgets while preserving new-task performance (Section 5).

# 2. Data-Free Constraints for Continual Learning

This section develops three ingredients for data-free continual learning: $(i)$ an exact invariance condition that yields zero forgetting but requires access to $P_0$, $(ii)$ a lower bound showing why approximate orthogonality admits a worst-case forgetting floor, and $(iii)$ a weight-only route to approximate protected subspaces via redundancy.

In this paper, we generally consider a two-task continual-learning setting. An old task is represented by a distribution $P_0$ over $(x, y)$ and a new task by $P_1$. A network $f_\theta$ is trained on $P_0$ to obtain parameters $\theta_0$, and then adapted on $P_1$ to obtain $\theta_1 = \theta_0 + \Delta\theta$. We define $L_0(\theta) := \mathbb{E}_{(x,y)\sim P_0}\big[\ell(f_\theta(x), y)\big]$ and $L_1(\theta) := \mathbb{E}_{(x,y)\sim P_1}\big[\ell(f_\theta(x), y)\big]$ as the old- and new-task losses, respectively. The forgetting on $P_0$ is $\mathcal{F}_0(\theta_0, \theta_1) := L_0(\theta_1) - L_0(\theta_0)$.

## 2.1. Layerwise exact invariance orthogonality

Let $h_\theta^{(\ell)}(x)$ denote the post-activation of layer $\ell$ and $z_\theta^{(\ell)}(x)$ its pre-activation. Ignoring biases for clarity

$$z_\theta^{(\ell)}(x) = W^{(\ell)} h_\theta^{(\ell-1)}(x), \qquad h_\theta^{(\ell)}(x) = \sigma\big(z_\theta^{(\ell)}(x)\big).$$

For an update $\Delta\theta = \{\Delta W^{(\ell)}\}_\ell$, we define a point-wise orthogonality condition on $P_0$.

**Definition 2.1** (Layerwise, per-sample orthogonality on $P_0$). We say that $\Delta\theta$ is *per-layer orthogonal on $P_0$* if for every layer $\ell$ and every $x \in \text{supp}(P_0)$

$$\Delta W^{(\ell)} h_{\theta_0}^{(\ell-1)}(x) = 0. \tag{1}$$

**Proposition 2.2** (Per-layer orthogonality yields no forgetting (Proof in Appendix A.1)). *If $\Delta\theta$ satisfies (1), then $\mathcal{F}_0(\theta_0, \theta_0 + \Delta\theta) = 0$.*

Proposition 2.2 is an exact invariance statement. Its limitation is practical: enforcing (1) requires access to old-task features $h_{\theta_0}^{(\ell-1)}(x)$ over $\text{supp}(P_0)$ and building an orthogonal complement (or projector) to their span, which is infeasible when the old distribution is unavailable and costly even when a replay buffer exists.

## 2.2. Approximate orthogonality implies a forgetting floor

In practice, orthogonality constraints are approximate. We quantify approximation using a first-order linearization and a distributional drift measure.

For small $\Delta\theta$, we use the following first-order linearization

$$f_{\theta_0 + \Delta\theta}(x) \approx f_{\theta_0}(x) + J_{\theta_0}(x)\Delta\theta,$$

where $J_{\theta_0}(x) := \nabla_\theta f_\theta(x)\big|_{\theta=\theta_0}$ is the Jacobian of the model output with respect to the parameters, evaluated at $\theta_0$. Thus $J_{\theta_0}(x)\Delta\theta$ gives the first-order change in the output at input $x$ induced by the parameter perturbation $\Delta\theta$.

To measure this drift under $x \sim P_0$, we define $\|J_{\theta_0}\Delta\theta\|_{L_2(P_0)} := \big(\mathbb{E}_{x\sim P_0}\big[\|J_{\theta_0}(x)\Delta\theta\|_2^2\big]\big)^{1/2}$. Now, If updates are restricted to a linear subspace $S \subset \mathbb{R}^{\dim(\theta)}$, we define the worst-case unit-norm drift radius as

$$\varepsilon(S) := \sup_{\|\Delta\theta\|_2=1, \ \Delta\theta \in S} \|J_{\theta_0}\Delta\theta\|_{L_2(P_0)}. \tag{2}$$

In particular, $\varepsilon(S) = 0$ means that $J_{\theta_0}(x)\Delta\theta = 0$ for all $x \in \text{supp}(P_0)$ and all $\Delta\theta \in S$, i.e., updates in $S$ induce no *first-order* output change on old inputs. This is a linearized analogue of the exact invariance in Proposition 2.2. Note that $\varepsilon(S)$ provides a quantitative measure of how "orthogonal" the update subspace $S$ is to the old-task features: it captures the worst-case first-order output drift on $P_0$ induced by a unit-norm update in $S$. Figure 1 provides an input-space visualization of this notion of drift: the heatmap tracks changes in the local input–output linearization around old and new data samples.

We now denote by $g_0 := \nabla_\theta L_0(\theta_0)$ and $H_0 := \nabla_\theta^2 L_0(\theta_0)$ the loss gradient and Hessian at $\theta_0$. Note that, for a model

that has been well-optimized on $P_0$, we typically have $g_0 \approx 0$, so small-step changes in $L_0$ are dominated by the quadratic term governed by $H_0$.

The following theorem shows that as soon as $\varepsilon(S) > 0$, e.g., as soon as orthogonality is only approximate, there always exists a direction inside $S$ that incurs a non-trivial amount of forgetting.

**Theorem 2.3** (Lower bound on worst-case forgetting under approximate orthogonality (Proof in Appendix A.2)). *Assume a curvature link between $H_0$ and $J_{\theta_0}$ (Assumption A.1 in Appendix A.2), then, there exist constants $c > 0$ and $\rho > 0$ such that $\exists \Delta\theta \in S$ with $\|\Delta\theta\|_2 = \rho$ for which*

$$\mathcal{F}_0(\theta_0, \theta_0 + \Delta\theta) \ \geq \ c\, \rho^2\, \varepsilon(S)^2 + O(\rho^3),$$

*where $c > 0$ depends only on local properties of $L_0$ around $\theta_0$, in particular the curvature-link in Assumption A.1.*

In other words, approximate protection leaves an unavoidable worst-case forgetting floor. As soon as $\varepsilon(S) > 0$, meaning the update family $S$ permits some nonzero first-order output drift on old inputs, there exists a direction in $S$ that increases the old-task loss by at least $\rho^2\varepsilon(S)^2$ for sufficiently small step size $\rho$.

### 2.3. Data-free protected subspaces from neuron redundancy

Orthogonality-based continual-learning methods estimate protected subspaces from old-task data (e.g., gradients or feature covariances) and constrain updates accordingly. In large pretrained models, the pretraining distribution $P_0$ is typically unavailable (and replay is often infeasible), motivating a *weight-only* proxy for old-feature directions.

Pretrained networks are highly redundant and compressible (Han et al., 2015a;b; Hoefler et al., 2021; Frankle & Carbin, 2018), implying that many neurons capture repeated directions. An explanation for why such repeated directions are tied to the data distribution is provided by deep neural collapse analyses. In particular, (Garrod & Keating, 2025) derive an explicit deep neural collapse structured solution form in a deep linear unconstrained-features model. A key consequence is that, at each layer, weight vectors are confined to a low-dimensional data-dependent subspace.

**Proposition 2.4** (Layerwise weights lie in a data-dependent prototype subspace (Garrod & Keating, 2025) (Proof in Appendix A.3)). *For each layer $\ell$ there exist prototype directions $\{\mu_c^{(\ell)}\}_{c=1}^K$ (per layer class-mean feature directions) such that every row $w_j^{(\ell)}$ of $W^{(\ell)}$ satisfies $w_j^{(\ell)} \in \text{span}\{\mu_1^{(\ell)}, \ldots, \mu_K^{(\ell)}\}$.*

Proposition 2.4 shows that at each layer, the trained weights concentrate in a low-dimensional subspace determined by the data. In a wide layer, this naturally leads to a large number of colinear weights within that subspace.

Motivated by Proposition 2.4, we treat densely repeated (high-cosine-similarity) neuron directions as a weight-only proxy for dominant pretraining-era feature directions. Concretely, by grouping colinear neurons and taking the orthogonal complement of their span, we obtain an update subspace intended to suppress interaction with the most salient old-task features as an approximation to Proposition 2.2. This yields a data-free route to an *approximately* protected update family with small (but generally nonzero) drift radius $\varepsilon(S)$, and therefore still inherits the nonzero worst-case floor from Theorem 2.3. In the next section we complement approximate input-side protection with an explicit drift control lens, and later instantiate it via redundant-neuron plasticity.

## 3. Low-Curvature Update Families

Theorem 2.3 shows that approximate orthogonality alone leaves a non-zero worst-case forgetting floor whenever $\varepsilon(S) > 0$. We now introduce a complementary geometric point of view: the *restricted curvature* of the old-task loss over an update family $S$. This allows us to derive upper bounds on worst case forgetting and understand how one can build an update subspace $S$ that will mitigate forgetting.

### 3.1. A local quadratic view of worst-case forgetting

We now consider the old-task loss $L_0$ around $\theta_0$

$$L_0(\theta_0 + \Delta\theta) \ \approx \ L_0(\theta_0) + g_0^\top \Delta\theta + \tfrac{1}{2}\, \Delta\theta^\top H_0 \Delta\theta,$$

where $g_0 = \nabla_\theta L_0(\theta_0)$ and $H_0 = \nabla_\theta^2 L_0(\theta_0)$. For a well-trained model, $g_0 \approx 0$ and the quadratic term dominates small-update forgetting, consistent with loss-landscape analyses that connect retention to the geometry of task optima and the widening effect of training regimes (Mirzadeh et al., 2020). For a linear update family subspace $S$, define its *restricted curvature*

$$\lambda(S) := \sup_{\Delta\theta \in S,\ \|\Delta\theta\|_2 = 1} \Delta\theta^\top H_0 \Delta\theta.$$

To capture worst-case behaviour, define the local worst-case forgetting over the update family subspace $S$

$$\mathcal{F}_{\max}(S, \rho) := \sup_{\substack{\Delta\theta \in S \\ \|\Delta\theta\|_2 \leq \rho}} \mathcal{F}_0(\theta_0, \theta_0 + \Delta\theta).$$

**Proposition 3.1** (Upper bound via restricted curvature (Proof in Appendix A.4)). *$\exists \rho > 0$ such that for any linear subspace $S$*

$$\mathcal{F}_{\max}(S, \rho) \leq \tfrac{\lambda(S)}{2}\rho^2 + O(\rho^3).$$

*In particular, for unconstrained full fine-tuning, $\mathcal{F}_{\max}(\mathbb{R}^{\dim(\theta)}, \rho) \leq \tfrac{\lambda_{\max}}{2}\rho^2 + O(\rho^3)$, where $\lambda_{\max}$ is the largest eigenvalue of $H_0$.*

Proposition 3.1 provides a design principle: to reduce worst-case forgetting, choose $S$ so that it *removes* high-curvature directions of the old loss. However, $\lambda(S)$ depends on the restriction of the old-task Hessian $H_0$ to $S$, which is generally intractable to compute and hard to approximate reliably without old-task data. We now show that under some assumptions, $\lambda(S)$ can be controlled by the *first-order* functional drift radius $\varepsilon(S)$ from Section 2.2.

### 3.2. From curvature to functional drift

We now connect the local worst-case forgetting control knob from Section 3.1, the *restricted curvature* $\lambda(S)$, to the *functional drift* that is directly targeted by orthogonality-style constraints. Recall the drift radius $\varepsilon(S)$ from Eq. 2 measures the worst-case *first-order* change in the model output on old-task inputs induced by a unit-norm update restricted to $S$.

For well-trained pretrained models, the old-task loss is typically close to stationary on the old distribution, so higher-order effects are weak and the curvature along an update direction is well captured by how much that update changes the model's outputs on old inputs.

**Proposition 3.2** (Restricted curvature is bounded by functional drift (Proof in Appendix A.5)). *Assume there exists $\beta > 0$ such that $\nabla_f^2 \ell(f_{\theta_0}(x), y) \preceq \beta I$, then for any linear subspace $S$*

$$\lambda(S) \;\leq\; \beta\, \varepsilon(S)^2.$$

Proposition 3.2 allows us to connect a second-order quantity $\lambda(S)$ on the loss into a first-order one $\varepsilon(S)$ on the network mapping. Specifically, it describes that if an update family produces small first-order output drift on $P_0$, then it cannot expose large old-task curvature.

Combining Proposition 3.1 (worst-case forgetting is controlled by $\lambda(S)$) with Proposition 3.2 (restricted curvature is controlled by $\varepsilon(S)$) yields the following upper bound on worst case forgetting.

**Theorem 3.3** (Worst-case forgetting is controlled by functional drift (Proof in Appendix A.7)). *There exist $\rho > 0$ such that for any linear subspace $S$*

$$\mathcal{F}_{\max}(S, \rho) \;\leq\; \frac{\beta}{2}\, \varepsilon(S)^2\, \rho^2 + O(\rho^3).$$

Theorem 3.3 shows that *to reduce local worst-case forgetting, it suffices to construct update families with small drift radius $\varepsilon(S)$ on $P_0$*. The remaining question is: how can we design such low-drift update families without access to $P_0$?

### 3.3. Designing low-drift update families without access to old-task data

The key ingredient to design such a low-drift update family without having to access the old-task data is redundancy in

pretrained layers: deep neural networks are highly compressible, and this redundancy is often expressed as repeated/colinear weight directions and near-duplicate features (Balestriero et al., 2021; 2019). We leverage this on the input side (approximate protection) and the output side (restricting plasticity to redundant channels):

1. **Input-side protection (weight-only approximate data-orthogonality).** Motivated by the weight-only construction in Section 2.3 and the drift-based design target in Section 3.3, we build an *approximately protected* input subspace using redundancy: we group highly colinear neurons (a proxy for dominant pretraining-era feature directions) and use its orthogonal subspace to suppress interaction with those directions. This mirrors orthogonality-style continual-learning constraints (Zeng et al., 2018; Farajtabar et al., 2019; Saha et al., 2021), but in a fully data-free, weight-only manner, leveraging redundancy and compressibility in trained networks.

2. **Output-side safety (redundant-channel restriction).** Motivated by the fact that approximate orthogonality alone admits a worst-case forgetting floor (Section 2.2) and by the drift/curvature control perspective (Sections 3.1-3.2), we further concentrate plasticity on a subset of *highly redundant* neurons. In fact, redundant neurons exhibit colinear hyperplanes and does not induced new partitions in the network input space. This biases updates toward degrees of freedom that are empirically shown to reduce the drift of the network mapping on the pre-trained data, consistent with geometric views introduced in (Balestriero et al., 2019; 2021) and broader evidence of redundancy in overparameterized trained networks (Han et al., 2015a;b; Frankle & Carbin, 2018; Hoefler et al., 2021).

Together, these two restrictions define a structured, low-dimensional update family that is explicitly designed to reduce old-task drift $\varepsilon(S)$ using only pretrained weights. We will see empirically (Section 5.3, with full sweeps in Appendix C.3) that these two ingredients play asymmetric roles: the input-side construction of $Q$ is the dominant driver of *retention* (replacing it with a random orthonormal basis at the same parameter count increases both old-distribution KL and Task-1 forgetting), while the output-side selection of $B$ primarily controls *where* the plasticity budget is spent and has only a mild effect on retention. Complementary restricted-curvature evidence is provided in Appendix C.2, Figure 11.

## 4. PLATE: Plasticity-Tunable Efficient Adapters

We now instantiate these principles in a parameter-efficient adapter-based method that is fully data-free with respect

to the old distribution $P_0$. The resulting method, PLATE (**Pla**sticity-**T**unable **E**fficient Adapters), constructs for each adapted linear map a structured update family that is designed to reduce $\varepsilon(S)$ using only frozen pretrained weights: $(i)$ it concentrates plasticity on a subset of *redundant* output channels and $(ii)$ it restricts updates to a *low-energy* input subspace inferred from the remaining frozen weights.

Specifically, for each layer $\ell$, PLATE defines a linear update family

$$S_{\text{PLATE}}^{(\ell)} := \{ B^{(\ell)} A^{(\ell)} Q^{(\ell)\top} \ : \ A^{(\ell)} \in \mathbb{R}^{r \times k} \},$$

where $B^{(\ell)} \in \mathbb{R}^{d_{\text{out}} \times r}$ is an *index-selection matrix* (a column submatrix of the identity) that selects $r$ trainable (redundant) output channels, $Q^{(\ell)} \in \mathbb{R}^{d_{\text{in}} \times k}$ spans a low-energy input subspace, and only $A^{(\ell)}$ is learned. Similar to LoRA and other PEFT methods, PLATE adds an adapter to the frozen modules that are selected for fine-tuning.

$$W' = W + \rho \Delta W, \qquad \Delta W = BAQ^\top.$$

PLATE adapter adds a rank$\leq \min(r, k)$ matrix to the existing frozen weight with $rk$ trainable parameters. The pseudo-code is described in Algorithm 1 and its computational complexity is provided in Appendix B.4 (Figure 8 for computational comparison with LoRA on DistilBERT fine-tuning).

## 4.1. Algorithm overview

PLATE consists of a one-time, weight-only preprocessing step per layer to compute $(B, Q)$, followed by standard adapter training on the new-task data with only $A$ trainable. The key point is that both ingredients are computed *once per model and without access to* $P_0$. In Figure 9 we provide details regarding the complexity of PLATE initialization.

PLATE exposes two main hyperparameters that provide an explicit trade-off between learning and forgetting: $(i)$ $r$ the number of adapted (redundant) output neurons, which controls the *plasticity budget*. Increasing $r$ typically improves learnability on the new task but can increase old-task forgetting as it increase the number of learnable neurons and therefore affect $\varepsilon(S)$, $(ii)$ $\tau$ an input energy threshold controlling the size of the orthogonal subspace, $k$. That is, $\tau$ controls how conservatively PLATE restricts updates to redundant degrees of freedom. Note that *larger* $\tau$ makes the constraint more stringent and induces a *smaller* $k$, because $k$ is chosen as the *smallest* dimension such that the complementary (high-energy) subspace captures a $\tau$ fraction of the spectrum/energy of $W_{\text{frozen}}$.

It is important to note that the additional flexibility PLATE offers in navigating the learning-forgetting trade-off comes at the cost of more involved engineering choices than simpler PEFT methods such as LoRA. Also, one should note

---

**Algorithm 1** PLATE

**Require:** Pretrained model parameters $\theta_0$, target linear layers $\mathcal{L}$, number of selected output neurons $r$, energy threshold for input subspace $\tau \in (0, 1)$, scaling $\rho$

1: **for** each layer $\ell \in \mathcal{L}$ with weight $W^{(\ell)} \in \mathbb{R}^{d_{\text{out}} \times d_{\text{in}}}$ **do**
2:     **Redundant-output selection:** choose indices $\mathcal{I}^{(\ell)}$ of cardinal $r$ using neuron similarity measure (Section B.1)
3:     Form $B^{(\ell)} = [e_i]_{i \in \mathcal{I}^{(\ell)}} \in \mathbb{R}^{d_{\text{out}} \times r}$, where $e_i$ denotes the $i$-th standard basis vector in $\mathbb{R}^{d_{\text{out}}}$
4:     Let $W_{\text{frozen}}^{(\ell)}$ be the submatrix of rows *not* in $\mathcal{I}^{(\ell)}$
5:     **Low-energy input basis:** compute $Q^{(\ell)} \in \mathbb{R}^{d_{\text{in}} \times k}$ from $W_{\text{frozen}}^{(\ell)}$ (Section B.2), where $k$ is chosen so the *complementary* high-energy subspace captures a $\tau$ fraction of the estimated energy
6:     Initialize trainable $A^{(\ell)} \in \mathbb{R}^{r \times k}$ (zero matrix)
7:     Define the adapted layer by $W'^{(\ell)} = W^{(\ell)} + \rho B^{(\ell)} A^{(\ell)} Q^{(\ell)\top}$
8: **end for**
9: Train only $\{A^{(\ell)}\}_{\ell \in \mathcal{L}}$ on the new-task data (all $W^{(\ell)}, B^{(\ell)}, Q^{(\ell)}$ frozen)

---

that the number of learnable parameters for LoRA and PLATE scale differently from the rank $r$. In fact, LoRA learnable parameters per layer are $2rd$ while for PLATE, there is no dependency on the number of hidden dimension, that is, the number of learnable parameters is $rk$. This is a crucial difference as in PLATE one can increase the rank while not drastically increasing the number of learnable parameters as the hidden dimension $d$ is usually large in LLMs.

## 5. Experiments

In this section, we provide the experimental results for PLATE. We evaluate PLATE in two complementary regimes:

1. **Out-of-distribution forgetting in LLMs**: we adapt pretrained LLMs on new reasoning/instruction-following dataset and probe both forgetting and learnability on *separate* benchmarks that were not used in training (Section 5.1). Training performed with open-instruct [1] and evaluation with OLMES (Gu et al., 2024).

2. **In-distribution forgetting in controlled benchmarks**: we build standard two-task continual-learning setups in vision, regression, and text modalities where the evaluation distribution coincides with a known training distribution (Section 5.2). For these experiments we follow the conventional two-stage continual-learning protocol described in Algorithm 3.

---

[1]https://github.com/allenai/open-instruct

For the out-of-distribution LLM experiments, we compare LoRA and PLATE. Across all in-distribution experiments, we compare Full FT (all weights trainable), LoRA, O-LoRA (Wang et al., 2023), L2-Init (Kumar et al., 2023), and PLATE. O-LoRA is a replay-free orthogonality-based PEFT method, while L2-Init regularizes all parameters toward their pre-task-2 values. Forgetting is harder to anticipate in OOD LLM experiments than in in-distribution settings: fine-tuning corpora overlap heterogeneously with the pretraining mixture, so it is rarely clear which capabilities are reinforced versus overwritten, and retention can shift qualitatively with the base model or optimization setup. We tune LoRA rank $r$ ($\alpha/r = 0.5$) and PLATE ($r, \tau$) at fixed $\rho = 0.5$ (LoRA-equivalent scaling); grids and architectures are in Tables 2 and 3.

Note that a condensed account of the assumptions and known failure modes (Gauss–Newton residual, weight-space vs. activation alignment, behavior under aggressive compression, and short-versus-long task streams) is provided in Appendix E.

### 5.1. Out-of-distribution forgetting in LLM specialization

We first consider the LLM setting where the pretraining distribution $P_0$ is unknown and inaccessible, and we evaluate forgetting on OOD benchmarks to analyze how much fine-tuning affect the generalization capabilities of the model.

#### 5.1.1. QWEN2.5-7B SPECIALIZED TO DEEPSEEK-R1 REASONING DATASET

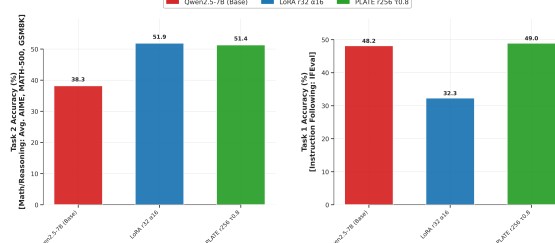

*Figure 2.* **Qwen2.5-7B on DeepSeek-R1 reasoning:** (**Left**) Learning capabilities on maths/reasoning dataset. (**Right**) Forgetting on instruction following dataset. PLATE (green) matches LoRA (blue) on math/reasoning benchmarks while preserving instruction-following (IFEval), whereas LoRA exhibits substantial OOD forgetting relative to the base model.

We start with Qwen2.5-7B (Yang et al., 2024a) and fine-tune on it the AM-DeepSeek-R1 distilled reasoning corpus (1 epoch, learning rate $10^{-4}$, AdamW). We compare: (1) base model, (2) LoRA (rank 32), and (3) PLATE (rank 256), adapters are attached to all modules. We evaluate (*i*) **OOD learning** on math/reasoning benchmarks (AIME, GSM8K, MATH-500); and (*ii*) **OOD forgetting** on IFEval, which probes instruction-following capabilities acquired during pretraining. Figure 2 shows that PLATE matches LoRA's $\approx$+13 point gain on math datasets while essentially eliminating the $\approx$16 point drop that LoRA incurs on IFEval.

#### 5.1.2. OLMO-2-7B SPECIALIZED TO TULU-3 DATASET

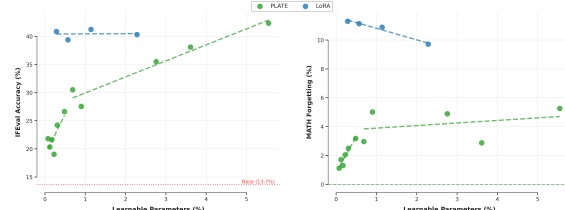

*Figure 3.* **OLMo-2-7B on Tulu-3:** (**Left**) IFEval accuracy vs. percentage of trainable parameters. The red dashed line is the base model capabilities on IFEval. (**Right**) MATH forgetting (drop from base) versus trainable parameters. PLATE (green) improves IFEval roughly linearly with parameter budget while keeping forgetting almost flat, whereas LoRA (blue) quickly saturates on IFEval and accumulates much larger MATH forgetting.

We then study how the percentage of learnable parameters affect both learning and forgetting capabilities. The base model is OLMo-2-7B (OLMo et al., 2024), we perform supervised fine-tuning on the Tulu-3 SFT OLMo-2 mixture (10% subsample, OLMo chat template, 1 epoch, learning rate $10^{-4}$, AdamW). We again attach adapters to all module. For LoRA we sweep $r \in \{8, 16, 32, 64\}$ and for PLATE we sweep the following configuration $r \in \{32, 128, 512, 1024\}$, $\tau \in \{0.8, 0.9, 0.95, 0.98\}$.

We measure **OOD learnability** as IFEval accuracy and **OOD forgetting** as the drop in MATH-500 performance relative to the base model. Although the Tulu-3 mixture contains reasoning and maths problems, MATH-500 is still severely affected by forgetting, reinforcing that OOD retention is hard to predict from the fine-tuning corpus alone. Figure 3 shows that PLATE can navigate the learning–forgetting trade-off by varying its number of learnable parameters, with forgetting nearly plateauing as that number grows; in the low-budget range LoRA outperforms PLATE on learnability. LoRA, by contrast, is relatively insensitive to its rank in this setting: performance changes only modestly across the sweep and forgetting remains hard to avoid, which explains its appeal as a robust default while also clarifying that meaningful control over the retention–plasticity trade-off requires methods like PLATE.

### 5.2. In-distribution forgetting benchmarks

We now consider settings where the task 1 distribution is known and forgetting can be measured directly on task 1 data after adapting to task 2.

#### 5.2.1. LANGUAGE MODELING: WIKITEXT-2 → MIDDLE ENGLISH

We now study continual adaptation in a language generation task. We start from a pretrained Qwen 2.5-3B model and fine-tune it on the Middle English dataset (EN-ME) for one epoch, comparing LoRA (varying rank) and PLATE (varying rank with fixed $\tau = 0.98$). We measure **learnability** as perplexity on EN-ME, and **retention** as perplexity on

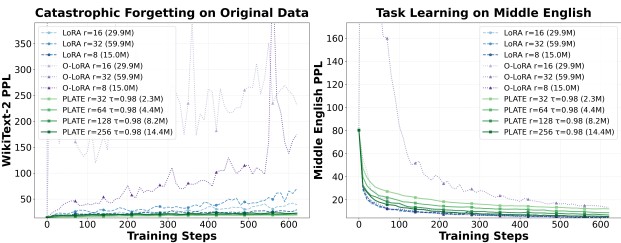

*Figure 4.* **Qwen 2.5-3B on Middle English (EN-ME):** Perplexity over training steps on WikiText-2 **(left: retention/forgetting)** and EN-ME **(right: learning)**. LoRA runs (blue, dashed) vary rank, while PLATE runs (green, solid) vary rank with $\tau = 0.98$. We observe again the capability of PLATE to alleviate catastrophic forgetting while allowing for adaptable learning regimes. Additional sweeps over $(r, \tau)$ are shown in Appendix C.7 Figure 17.

WikiText-2 as a proxy for general-domain pretraining behavior. Figure 4 summarizes the resulting learning–retention trade-off, again showing that PLATE consistently mitigates catastrophic forgetting, while LoRA offers only limited control over retention in this setting. The orthogonality-based O-LoRA baseline, despite sharing the same parameter budget as LoRA at every rank, exhibits 10–100× larger forgetting on WikiText-2: under this large domain shift the orthogonal re-initialization actively destabilizes the original representations rather than protecting them. L2-Init diverged across all regularization strengths $\lambda$ we tested on the 3B model, alternating between catastrophic forgetting and failure to learn the new task, and is therefore omitted from the figure. Additional sweeps over $(r, \tau)$ allowing to better capture the retention–plasticity spectrum PLATE provides are shown in Figure 17. We now turn to controlled settings where the task 1 distribution is known and forgetting can be measured exactly. For these experiments, we train a randomly initialized model on task 1, and then fine-tune it on task 2 (Table 3).

### 5.2.2. VISION: MNIST 0-4 → 5-9

For this experiment, task 1 is a classification problem on MNIST digits $\{0, \ldots, 4\}$, and task 2 is the classification on MNIST digits $\{5, \ldots, 9\}$ using a shared 3-layer ReLU MLP backbone (Table 3). We sweep LoRA ranks $r \in \{1, 8, 16, 32, 64, 128\}$ and PLATE ranks $r \in \{32, 64, 128, 256, 350\}$ (with $\tau = 0.8$), all applied to the backbone layers. Figure 5 (top) summarizes the results with respect to the % of learnable parameters. All methods reach $\approx 98\%$ task 2 accuracy once they have at least a few percent of the backbone parameters as trainable. The key difference lies in task 1 retention: full fine-tuning forgets about 26% of task 1 accuracy despite excellent task 2 performance; LoRA reduces forgetting to $\approx 7\text{-}9\%$. PLATE, by contrast, achieves near-full retention: at $10.2\%$ trainable parameters it reaches $98.28\%$ task 2 accuracy and $97.45\%$ task 1 retention, i.e. only $1.85\%$ forgetting, more than $4\times$ better than LoRA at similar capacity. The two replay-free baselines occupy

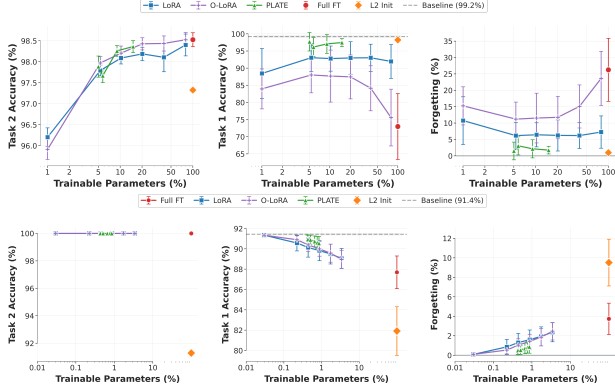

*Figure 5.* **In distribution Vision and text benchmarks:** (**Top**): MNIST 0-4→5-9, showing task 2 performance and task 1 forgetting as a function of trainable parameters. (**Bottom**): AG News→IMDB, with all methods achieving near-perfect task 2 accuracy while differing in how much they forget task 1.

opposite ends of the trade-off: L2-Init matches PLATE's task-1 retention ($\sim 98\%$) but only at $100\%$ trainable parameters (no parameter efficiency), while O-LoRA, despite sharing the same per-rank budget as LoRA, consistently forgets more, with retention degrading from $\sim 88\%$ at $r=8$ down to $\sim 76\%$ at $r=128$ and converging toward full fine-tuning behavior as rank grows. PLATE is the only method that simultaneously achieves near-full retention and high parameter efficiency.

### 5.2.3. TEXT CLASSIFICATION: AG NEWS → IMDB

Finally we revisit the text modality in the *in-distribution* sense: we know both training and evaluation distributions for AG News (task 1) and IMDB (task 2) and can measure forgetting exactly. We pretrain DistilBERT-base (Sanh et al., 2019) on AG News (3 epochs, learning rate $2 \times 10^{-5}$) obtaining a baseline task 1 accuracy of $91.34\% \pm 0.12\%$, then adapt to IMDB with: full fine-tuning; LoRA with $r \in \{1, 8, 16, 32, 64, 128\}$; and PLATE with fixed rank $r = 32$, thresholds $\tau \in \{0.6, 0.7, 0.8, 0.9\}$.

Figure 5 (bottom) shows that all methods reach $100\%$ IMDB accuracy, so the trade-off is purely about task 1 retention. Full fine-tuning incurs about $3\%$ forgetting. LoRA displays a rank-dependent trade-off: a rank-1 configuration achieves zero forgetting, but forgetting rises steadily to $\approx 2\text{-}3\%$ as rank increases. PLATE, by contrast, keeps forgetting below $0.5\%$ across all configurations. On this "free lunch" dataset where all adapter methods reach $100\%$ task-2 accuracy, L2-Init is the only baseline that fails to learn task 2 (capped at $\sim 91.3\%$) while also suffering the worst task-1 retention, indicating that its uniform regularization is too rigid for the AG News→IMDB distribution shift. O-LoRA tracks LoRA closely on this dataset, suggesting that the orthogonal-update prior provides no advantage in the small-shift regime. Note that additional experiments are provided in Appendix C.

## 5.3. Mechanism analysis: direct drift control and the role of $B$ vs $Q$

The previous experiments confirm that PLATE retains better than LoRA and the replay-free baselines, but they do not directly verify the geometric mechanism predicted by our analysis. Theorem 3.3 states that worst-case forgetting is controlled by the functional drift radius $\varepsilon(S)$ on $P_0$; our two-side construction is designed to make $\varepsilon(S)$ small in a data-free way, with the input-side basis $Q$ as the dominant lever and the output-side selector $B$ controlling where plasticity is spent. We now verify both claims directly.

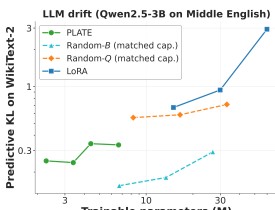 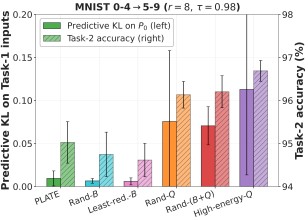

*Figure 6.* **Direct evidence for the drift-based mechanism. (Left)** Predictive KL between the base and adapted Qwen2.5-3B on WikiText-2 inputs as a function of trainable parameters, after Middle English adaptation. PLATE (green) achieves the lowest KL across the entire parameter range; LoRA exhibits up to an order of magnitude larger drift at comparable or larger budgets. Matched-capacity Random-$Q$ adapters (orange) shift PLATE upward by 2-3$\times$, while matched-capacity Random-$B$ adapters (cyan) track PLATE closely. **(Right)** Matched-capacity controls on MNIST 0-4$\rightarrow$5-9 at the canonical config ($r{=}8, \tau{=}0.98$), averaged over 5 seeds: predictive KL on Task-1 inputs (solid, left axis) and Task-2 accuracy (hatched, right axis). Randomizing $Q$ (Rand-$Q$, Rand-$(B{+}Q)$, High-energy-$Q$) drives the KL up by 8-12$\times$. Among the $Q$-preserving variants, PLATE's redundant-channel $B$ yields the highest Task-2 accuracy, confirming that $B$ controls *where* the plasticity budget is spent even though it does not affect retention.

**Direct drift evidence (LLM).** Figure 6 (left) measures the symmetrized KL between the base and adapted Qwen2.5-3B, averaged token-wise over the WikiText-2 evaluation set; for the MNIST panel (right) the same quantity is computed on the model's class logits over Task-1 inputs. This output-space drift is a legitimate empirical proxy for $\varepsilon(S)$: to leading order in $\Delta\theta$, predictive KL scales with $\|J_{\theta_0}\Delta\theta\|^2_{L_2(P_0)}$, so a small KL on the pretraining-era inputs directly bounds the worst-case forgetting via Theorem 3.3. PLATE reaches KL $\approx 0.24$ with 3.4M trainable parameters; LoRA requires $4{-}10\times$ more parameters and still exhibits $3{-}10\times$ larger KL. This is a direct functional translation of the perplexity results in Section 5.2.1: PLATE's retention gain on WikiText-2 reflects measurably smaller output drift on the old distribution.

**Mechanism: $Q$ drives retention, $B$ allocates plasticity.** Figure 6 (right) isolates each of PLATE's two design choices via matched-capacity controls. Replacing the weight-derived input basis $Q$ by a random orthonormal $k$-frame (Rand-$Q$) or by the top (rather than bottom) eigen-

vectors of $G_{\mathrm{in}}$ (High-energy-$Q$) increases the predictive KL on $P_0$ from 0.009 to $0.076{-}0.113$ ($8{-}12\times$) and the Task-1 forgetting from $0.20\%$ to $1.5{-}2.4\%$, while keeping every other element of the architecture and parameter count identical. In contrast, replacing the redundant-output selector $B$ by a uniformly random subset (Rand-$B$) or by the *least* redundant outputs (Least-red.-$B$) leaves the KL and Task-1 forgetting essentially unchanged ($0.006{-}0.007$ and $\approx 0.19\%$, respectively), but it does have a small and consistent effect on *new-task* learnability: at the same fixed $Q$ and parameter count, PLATE's redundant-channel $B$ reaches $95.03\%$ Task-2 accuracy versus $94.75\%$ for Rand-$B$ and $94.61\%$ for Least-red.-$B$. The same asymmetry is visible on the LLM side: Random-$Q$ shifts the drift curve substantially upward at every parameter budget, while Random-$B$ stays close to PLATE. This empirically separates the two roles predicted by the theory: the input-side construction of $Q$ is the dominant driver of retention (it controls $\varepsilon(S)$), while the output-side selection of $B$ controls how the plasticity budget is allocated, it does not affect retention, but it does mildly improve Task-2 adaptation. Extended ablations (full sweep, additional controls, and per-layer breakdown) are provided in Appendix C.3.

## 6. Discussion and Conclusion

We studied catastrophic forgetting in parameter-efficient adaptation of foundation models in the practically relevant regime where the pretraining distribution $P_0$ is unavailable and replay is impractical. Our analysis identifies *functional drift on old inputs* as the key geometric quantity governing worst-case forgetting: if an update family permits nonzero drift on $P_0$, it necessarily contains directions that incur a nontrivial forgetting floor, while controlling drift also controls restricted curvature and yields an upper bound on worst-case forgetting. This suggests a simple design principle: build update families that keep output drift on $P_0$ small. We instantiate this principle with PLATE, a structured PEFT adapter $\Delta W = BAQ^\top$ that is fully data-free with respect to $P_0$: it restricts updates to a weight-derived low-energy input subspace and concentrates plasticity on redundant output channels. LoRA remains appealing as a simple and robust default; however, when retention is a first-class constraint, methods with finer control over forgetting are needed. PLATE is designed for this setting: it matches LoRA on new-task gains while improving retention across continual-learning benchmarks and LLM specialization, and it exposes explicit retention–plasticity control via $(r, \tau)$. More broadly, the drift–curvature link of Theorem 3.3, together with its empirical validation in Section 5.3, suggests a method-agnostic evaluation lens: any new continual-learning method, whether it modifies $B$, $Q$, or neither, can be assessed by how well it controls $\varepsilon(S)$ on the old distribution, independently of its parameter count or architectural family.

## Impact Statement

This paper presents work whose goal is to advance the field of Machine Learning. There are many potential societal consequences of our work, none which we feel must be specifically highlighted here.

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

# A. Proofs

## A.1. Proof of Proposition 2.2

*Proof.* Recall that for each layer $\ell$ we have

$$z_\theta^{(\ell)}(x) = W^{(\ell)} h_\theta^{(\ell-1)}(x), \qquad h_\theta^{(\ell)}(x) = \sigma(z_\theta^{(\ell)}(x)),$$

and that $\theta_1 = \theta_0 + \Delta\theta$ with $\Delta\theta = \{\Delta W^{(\ell)}\}_\ell$.
We prove by induction on $\ell$ that for every $x \in \text{supp}(P_0)$,

$$h_{\theta_1}^{(\ell)}(x) = h_{\theta_0}^{(\ell)}(x).$$

**Base case ($\ell = 1$).** By definition,

$$z_{\theta_1}^{(1)}(x) = (W^{(1)} + \Delta W^{(1)}) h_{\theta_1}^{(0)}(x) = (W^{(1)} + \Delta W^{(1)}) x.$$

Using the per-neuron orthogonality condition (1) with $h_{\theta_0}^{(0)}(x) = x$ gives $\Delta W^{(1)} x = 0$, hence

$$z_{\theta_1}^{(1)}(x) = W^{(1)} x = z_{\theta_0}^{(1)}(x),$$

and therefore $h_{\theta_1}^{(1)}(x) = h_{\theta_0}^{(1)}(x)$.

**Induction step.** Assume that for some $\ell \geq 1$ we have $h_{\theta_1}^{(\ell-1)}(x) = h_{\theta_0}^{(\ell-1)}(x)$ for all $x \in \text{supp}(P_0)$. Then

$$\begin{aligned}
z_{\theta_1}^{(\ell)}(x) &= (W^{(\ell)} + \Delta W^{(\ell)}) h_{\theta_1}^{(\ell-1)}(x) \\
&= (W^{(\ell)} + \Delta W^{(\ell)}) h_{\theta_0}^{(\ell-1)}(x) \\
&= W^{(\ell)} h_{\theta_0}^{(\ell-1)}(x) + \Delta W^{(\ell)} h_{\theta_0}^{(\ell-1)}(x).
\end{aligned}$$

By the per-neuron orthogonality condition (1), $\Delta W^{(\ell)} h_{\theta_0}^{(\ell-1)}(x) = 0$ for all $(x, y) \sim P_0$, so

$$z_{\theta_1}^{(\ell)}(x) = W^{(\ell)} h_{\theta_0}^{(\ell-1)}(x) = z_{\theta_0}^{(\ell)}(x).$$

Thus $h_{\theta_1}^{(\ell)}(x) = h_{\theta_0}^{(\ell)}(x)$.
In particular, the final layer outputs coincide for all $x \in \text{supp}(P_0)$:

$$f_{\theta_1}(x) = f_{\theta_0}(x),$$

which implies $L_0(\theta_1) = L_0(\theta_0)$ and therefore $\mathcal{F}_0(\theta_0, \theta_1) = 0$. $\qquad\square$

## A.2. Curvature assumption and proof of Theorem 2.3

**Assumption A.1** (Output-space curvature link). There exists a constant $\mu_0 > 0$ such that for all $\Delta\theta \in S$,

$$\Delta\theta^\top H_0 \Delta\theta \geq \mu_0 \, \mathbb{E}_{x \sim P_0}\left[\|J_{\theta_0}(x)\Delta\theta\|_2^2\right]. \tag{3}$$

Intuitively, directions in parameter space that induce a large first-order change in the network output on $P_0$ must incur a proportional amount of curvature in the old-task loss $L_0$.
Recall the definition of $\varepsilon(S)$ from (2)

$$\varepsilon(S) := \sup_{\substack{\Delta\theta \in S \\ \|\Delta\theta\|_2 = 1}} \left(\mathbb{E}_{x \sim P_0}\left[\|J_{\theta_0}(x)\Delta\theta\|_2^2\right]\right)^{1/2}. \tag{4}$$

*Proof.* Define

$$\Phi(\Delta\theta) := \left(\mathbb{E}_{x \sim P_0}\left[\|J_{\theta_0}(x)\Delta\theta\|_2^2\right]\right)^{1/2}.$$

$\Phi$ is continuous and the feasible set $\{\Delta\theta \in S : \|\Delta\theta\|_2 = 1\}$ is compact, so the supremum in Eq.2 is attained. Thus there exists $\Delta\theta^\star \in S$ with $\|\Delta\theta^\star\|_2 = 1$ such that

$$\Phi(\Delta\theta^\star) = \varepsilon(S). \tag{5}$$

For a given step size $\rho > 0$, consider the scaled update $\Delta\theta_\rho := \rho\,\Delta\theta^\star \in S$ so that $\|\Delta\theta_\rho\|_2 = \rho$. We have,

$$\mathbb{E}_{x\sim P_0}\left[\|J_{\theta_0}(x)\Delta\theta_\rho\|_2^2\right] = \rho^2\,\mathbb{E}_{x\sim P_0}\left[\|J_{\theta_0}(x)\Delta\theta^\star\|_2^2\right] = \rho^2\,\varepsilon(S)^2.$$

Applying Assumption A.1 to $\Delta\theta_\rho$ yields

$$\Delta\theta_\rho^\top H_0 \Delta\theta_\rho \;\geq\; \mu_0\,\mathbb{E}_{x\sim P_0}\left[\|J_{\theta_0}(x)\Delta\theta_\rho\|_2^2\right] = \mu_0\,\rho^2\varepsilon(S)^2. \tag{6}$$

Using the Taylor expansion with $\Delta\theta = \Delta\theta_\rho$ and $g_0 \approx 0$ gives

$$\begin{aligned}
\mathcal{F}_0(\theta_0, \theta_0 + \Delta\theta_\rho) &= L_0(\theta_0 + \Delta\theta_\rho) - L_0(\theta_0) \\
&= g_0^\top \Delta\theta_\rho + \frac{1}{2}\,\Delta\theta_\rho^\top H_0 \Delta\theta_\rho + R(\Delta\theta_\rho).
\end{aligned} \tag{7}$$

For a well-trained model we take $g_0 \approx 0$ and neglect the linear term. We obtain

$$\begin{aligned}
\mathcal{F}_0(\theta_0, \theta_0 + \Delta\theta_\rho) &\geq \frac{1}{2}\,\mu_0\rho^2\varepsilon(S)^2 - C_T\rho^3 \\
&= \left(\frac{\mu_0}{2}\right)\rho^2\varepsilon(S)^2 - C_T\rho^3.
\end{aligned}$$

$\square$

### A.3. Proof of Proposition 2.4

*Proof.* This follows directly from the explicit Deep Neural Collapse solution form derived in (Garrod & Keating, 2025), where each row direction is expressed (up to scale) as a linear combination of the layerwise prototype directions; see the displayed derivation yielding $w_j^{(\ell)} \propto \sum_{c=1}^{K} O_{cj}\mu_c^{(\ell)}$. $\square$

### A.4. Proof of Proposition 3.1

*Proof.* For any $\Delta\theta \in S$ with $\|\Delta\theta\| \leq \rho$, since we have $g_0 = \nabla_\theta L_0(\theta_0) \approx 0$

$$\mathcal{F}_0(\theta_0, \theta_0 + \Delta\theta) = \frac{1}{2}\Delta\theta^\top H_0 \Delta\theta + O(\Delta\theta^3)$$

By definition of $\lambda(S)$, $\Delta\theta^\top H_0 \Delta\theta \leq \lambda(S)\|\Delta\theta\|^2 \leq \lambda(S)\rho^2$. Thus

$$\mathcal{F}_0(\theta_0, \theta_0 + \Delta\theta) \leq \frac{\lambda(S)}{2}\rho^2 + O(\rho^3).$$

The unconstrained case follows from $\lambda(\mathbb{R}^{\dim(\theta)}) = \lambda_{\max}$. $\square$

### A.5. Proof of Proposition 3.2

*Proof.* Recall that

$$L_0(\theta) = \mathbb{E}_{(x,y)\sim P_0}\left[\ell(f_\theta(x), y)\right], \qquad H_0 = \nabla_\theta^2 L_0(\theta_0) = \mathbb{E}_{(x,y)\sim P_0}\left[\nabla_\theta^2 \ell(f_\theta(x), y)\right]_{\theta=\theta_0}.$$

Fix $(x, y)$ and define $\phi(\theta) := \ell(f_\theta(x), y)$. Then

$$\nabla_\theta^2 \phi(\theta) = J_\theta(x)^\top \nabla_f^2 \ell(f_\theta(x), y)\, J_\theta(x) \;+\; \sum_{i=1}^{d_{\text{out}}} \frac{\partial \ell}{\partial f_i}(f_\theta(x), y)\, \nabla_\theta^2 f_{\theta,i}(x), \tag{8}$$

where $J_\theta(x) = \nabla_\theta f_\theta(x)$ and $f_{\theta,i}(x)$ denotes the $i$-th output coordinate.

The second term in (8) is the only deviation from Gauss–Newton curvature, and it is weighted by the loss sensitivity to the model output. For a well-optimized pretrained model on the old distribution, this output sensitivity is small on old-task data, so we treat the second term as a residual curvature contribution. Concretely, we assume that along the update directions of

interest this residual contribution is uniformly bounded by a constant $R$ (and in the idealized case of zero loss on $P_0$, it vanishes). For analytical clarity we present the bound with $R = 0$; empirically the residual is not exactly zero, but as we show in Appendix A.6 the Gauss–Newton component is substantially more visible inside the PLATE subspace than inside generic backbone or LoRA directions, making the bound below an informative design principle for PLATE in particular. Now, using $\nabla^2_f \ell(f_{\theta_0}(x), y) \preceq \beta I$ we have for any vector $v$

$$v^\top \nabla^2_\theta \ell(f_{\theta_0}(x), y)\, v = (J_{\theta_0}(x)v)^\top \nabla^2_f \ell(f_{\theta_0}(x), y)\, (J_{\theta_0}(x)v) \;\leq\; \beta \left\| J_{\theta_0}(x)v \right\|_2^2.$$

Taking expectation over $(x, y) \sim P_0$ gives

$$v^\top H_0 v \;=\; \mathbb{E}_{(x,y) \sim P_0}\left[ v^\top \nabla^2_\theta \ell(f_{\theta_0}(x), y)\, v \right] \;\leq\; \beta\, \mathbb{E}_{x \sim P_0}\left[ \| J_{\theta_0}(x)v \|_2^2 \right].$$

Now restrict to $v \in S$ with $\|v\|_2 = 1$ and take the supremum

$$\lambda(S) = \sup_{\substack{v \in S \\ \|v\|_2 = 1}} v^\top H_0 v \;\leq\; \beta \sup_{\substack{v \in S \\ \|v\|_2 = 1}} \mathbb{E}_{x \sim P_0}\left[ \| J_{\theta_0}(x)v \|_2^2 \right] = \beta\, \varepsilon(S)^2.$$

$\square$

### A.6. Empirical diagnostic of the Gauss–Newton decomposition

Proposition 3.2 discards the residual second-order term in Eq. (8) for clarity. We test the regime in which this approximation is most informative by directly measuring, at the old-task solution $\theta_0$, the directional decomposition

$$q_H(v) \;=\; v^\top H_0 v, \qquad q_{GN}(v) \;=\; v^\top J_{\theta_0}^\top \nabla^2_f \ell\, J_{\theta_0}\, v, \qquad q_R(v) \;=\; q_H(v) - q_{GN}(v),$$

for unit-norm directions $v$ drawn from three update families: the full backbone (generic), the LoRA adapter subspace, and the PLATE subspace. We sample directions across multiple seeds and batches and aggregate the resulting 1,890 probes ($\sim 600$ per family).

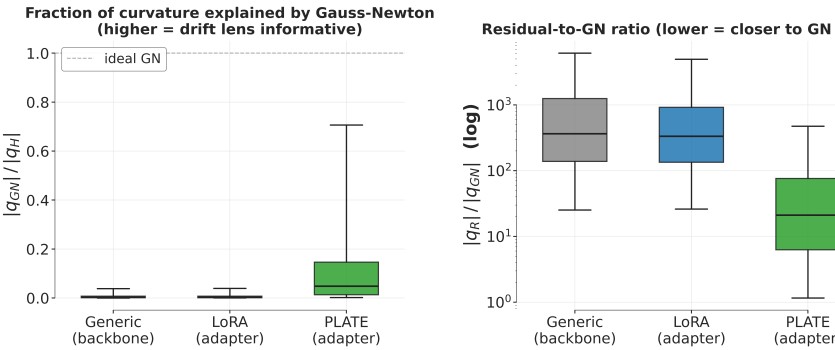

*Figure 7.* **Curvature decomposition at $\theta_0$ on MNIST. (Left)** Fraction of directional curvature explained by the Gauss–Newton term ($|q_{GN}|/|q_H|$); higher means the drift-based lens of Section 3.2 is more informative. **(Right)** Residual-to-Gauss–Newton ratio ($|q_R|/|q_{GN}|$) on log scale; lower means closer to the pure Gauss–Newton regime. Across $\sim 1{,}890$ directional probes, generic backbone and LoRA directions are dominated by the residual term (median $|q_{GN}|/|q_H| \approx 0.003$, $|q_R|/|q_{GN}| \approx 350$). PLATE directions are the only ones for which the Gauss–Newton component becomes materially visible (median $|q_{GN}|/|q_H| \approx 0.05$, $|q_R|/|q_{GN}| \approx 21$), more than an order of magnitude closer to the GN regime than the other tested families.

The residual term does not vanish along PLATE directions, but PLATE is the only tested update family in which the Gauss–Newton component becomes a non-negligible fraction of the total curvature. This is consistent with PLATE's geometric construction: by concentrating updates on redundant input/output directions, PLATE selects a subspace where small functional drift on $P_0$ *actually* translates into small curvature, confirming how informative is the drift-based bound of Proposition 3.2.

### A.7. Proof of Theorem 3.3

*Proof.* By Proposition 3.1, there exist $\rho > 0$ such that for any linear subspace $S$

$$\mathcal{F}_{\max}(S, \rho) \;\leq\; \frac{\lambda(S)}{2} \rho^2 + C\rho^3.$$

Under $\nabla_f^2 \ell(f_{\theta_0}(x), y) \preceq \beta I$, Proposition 3.2 yields $\lambda(S) \leq \beta \, \varepsilon(S)^2$. Therefore

$$\mathcal{F}_{\max}(S, \rho) \; \leq \; \frac{\beta}{2} \, \varepsilon(S)^2 \, \rho^2 + O(\rho^3),$$

$\square$

## B. Algorithms & Computational Details

### B.1. Constructing the redundant-neuron selector $B$

The first mechanism to reduce drift is to restrict plasticity to *redundant* output channels (Section 3.3). Intuitively, if a layer's output direction is implemented repeatedly by many other neurons, then modifying one instance tends to induce smaller functional change on the pretrained representation, and reduces drift on old inputs (Section 2.3).

As theoretically justified for pruning in (Balestriero et al., 2021), PLATE selects trainable output neurons using a simple redundancy heuristic: output rows that are highly colinear with many others are treated as redundant and are therefore candidates for plasticity (see also Section 3.3). Concretely, for each layer we score each output neuron $w_i^\top$ of $W$ by an estimate of its average cosine similarity to a set of *anchor* rows, and we pick the top-$r$ rows using this score.

To scale to large $d_{\text{in}}$, we compute similarities in a random projection space. Let $R \in \mathbb{R}^{d_{\text{in}} \times d'}$ be a random Gaussian projection (with $d' \ll d_{\text{in}}$) and define

$$z_i \; = \; \frac{w_i^\top R}{\|w_i\|_2} \in \mathbb{R}^{d'}.$$

We choose a set of anchor indices $\mathcal{A}$, and score each neuron by

$$s_i \; = \; \frac{1}{|\mathcal{A}|} \sum_{j \in \mathcal{A}} |\langle z_i, z_j \rangle|. \tag{9}$$

PLATE sets $\mathcal{I}$ to the indices of the $r$ largest scores $\{s_i\}$ and defines $B = [e_i]_{i \in \mathcal{I}}$.

Rows with high $s_i$ lie in densely populated directions of row space (directions implemented repeatedly by many neurons). Concentrating plasticity on these rows is therefore a pretrained distribution data-free bias toward update directions that induce smaller functional drift on the pretrained behavior.

### B.2. Constructing the weight-derived input basis $Q$

The second mechanism to reduce drift is to constrain the input side of the update to directions that are "low-energy" with respect to the frozen part of the layer as a proxy to "low-energy" $P_0$ data directions . The design choice is that $Q$ is computed from the *complement* of the selected trainable rows.

Let $\mathcal{I}$ be the selected indices and define the frozen-row submatrix

$$W_{\text{frozen}} \; \in \; \mathbb{R}^{(d_{\text{out}} - r) \times d_{\text{in}}} \quad \text{by removing rows indexed by } \mathcal{I},$$

and consider its Gram matrix $G_{\text{in}} := W_{\text{frozen}}^\top W_{\text{frozen}} \in \mathbb{R}^{d_{\text{in}} \times d_{\text{in}}}$. This low-energy (bottom-eigenspace) construction is the same weight-only protected input subspace described in Section 2.3: it corresponds to the (approximate) nullspace of $W_{\text{frozen}}$, i.e., directions orthogonal to the span of frozen neurons (our proxy for dominant pretraining-era feature directions). Directions with small quadratic form under $G_{\text{in}}$ correspond to inputs that weakly excite the frozen neurons. PLATE defines $Q$ as an orthonormal basis spanning a *low-energy* subspace of $G_{\text{in}}$, i.e., $Q$ approximates the bottom-eigenspace of $G_{\text{in}}$. Restricting updates to this subspace is intended to limit interaction with dominant pretrained features, and thus reduce first-order output drift on old inputs.

PLATE selects the basis dimension $k$ using a threshold $\tau \in (0, 1)$: $k$ is chosen so that the complementary (high-energy) subspace captures approximately a $\tau$ fraction of the estimated energy, and $k$ is capped by $k_{\max}$. This yields an explicit *plasticity knob*: larger $\tau$ (inducing smaller $k$) enforces stricter input constraints and typically improves retention.

When $d_{\text{in}}$ is large, PLATE avoids forming $G_{\text{in}}$ explicitly and instead uses a structured randomized Hadamard transform (SRHT) together with batched Hutchinson-style probes to estimate low-energy directions efficiently. Concretely, PLATE (i) applies an SRHT rotation, (ii) screens candidate coordinates using coarse then refined probes, and (iii) polishes within the candidate span by solving a small eigenproblem to recover a $k$-dimensional low-energy subspace.

### B.3. Sequential PLATE for $T \geq 3$ tasks

Algorithm 1 specifies PLATE for the two-task setting that is the focus of the main body. For task streams of length $T \geq 3$, we use a simple recursive extension: after each task, the trained adapter is merged into the backbone and the PLATE preprocessing is recomputed on the updated weights, refreshing the protected and plastic subspaces for the next task.

---

**Algorithm 2** Sequential PLATE for $T \geq 3$ tasks

---

**Require:** Pretrained model parameters $\theta_0$ with weights $\{W_0^{(\ell)}\}_\ell$, task stream $\mathcal{D}_1, \ldots, \mathcal{D}_T$, hyperparameters $(r, \tau, \rho)$
 1: **for** $t = 1$ to $T$ **do**
 2:    Run PLATE preprocessing (Algorithm 1, lines 2–6) on the current backbone $\{W_{t-1}^{(\ell)}\}_\ell$ to obtain layerwise selectors $\{B_t^{(\ell)}\}$, low-energy input bases $\{Q_t^{(\ell)}\}$, and freshly initialized adapter cores $\{A_t^{(\ell)}\}$
 3:    Train only $\{A_t^{(\ell)}\}_\ell$ (and the task-$t$ head) on $\mathcal{D}_t$
 4:    Merge the trained adapter into the backbone, $W_t^{(\ell)} \leftarrow W_{t-1}^{(\ell)} + \rho\, B_t^{(\ell)} A_t^{(\ell)} Q_t^{(\ell)\top}$ for all $\ell$
 5:    Save the task-$t$ head for inference on $\mathcal{D}_t$
 6: **end for**

---

Two properties of this recursion are worth highlighting. First, the protected and plastic subspaces $(B_t, Q_t)$ are *recomputed* after each merge from the most recent backbone, so PLATE is not tied to one fixed input nullspace: the admissible update family adapts to the evolving model geometry. Second, every per-task update remains weight-only (no access to $\mathcal{D}_1, \ldots, \mathcal{D}_{t-1}$ is required at task $t$). The cost of this construction is that the redundancy PLATE exploits is shared across tasks: in principle, the available pool can shrink as updates accumulate. We empirically characterize this behaviour over four sequential MNIST tasks in Appendix C.6, where retention degrades mildly ($\sim 6\%$ on Task 1 after four merges at the canonical config) and the $\tau$-thresholded pool remains stable over the horizon we test.

### B.4. Computational Complexity Analysis

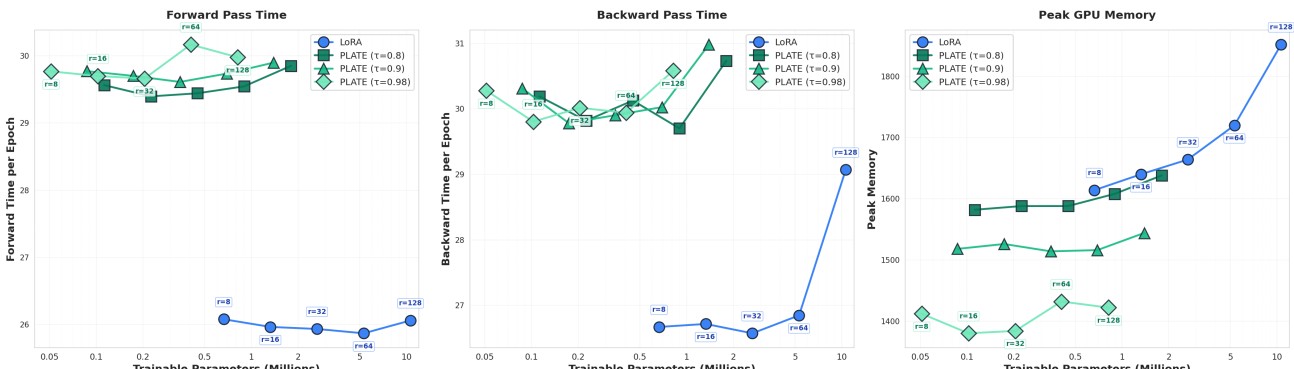

*Figure 8.* **Training efficiency of PLATE vs. LoRA on DistilBERT.** We adapt all linear layers and measure forward/backward time per epoch and peak GPU memory as a function of the total number of *trainable* adapter parameters. $(i)$ For a fixed output rank $r$, PLATE trains only $A \in \mathbb{R}^{r \times k}$ and therefore uses $rk$ trainable parameters per layer, whereas LoRA trains two matrices and uses $r(d_{\text{in}} + d_{\text{out}})$; across the sweep this yields fewer trainable parameters for PLATE, reducing optimizer-state and checkpoint size. $(ii)$ Despite storing frozen bases $Q$, PLATE achieves lower peak GPU memory than LoRA in this setting, due to reduced optimizer states and a smaller adapter-induced activation footprint (only $A$ is trainable). $(iii)$ PLATE incurs a $\sim 10\%$–$15\%$ per-epoch time overhead, driven primarily by the extra projection through $Q$. $(iv)$ Lowering $\tau$ (darker green) increases the induced basis dimension $k$ and therefore increases memory, while leaving training time nearly unchanged. Overall, PLATE trades a modest compute overhead for substantially fewer trainable parameters, improved memory efficiency, and a trade-off between plasticity and memory retention.

We compare the additional compute and memory introduced by the adapter branch of LoRA and PLATE for a linear layer $W \in \mathbb{R}^{d_{\text{out}} \times d_{\text{in}}}$ and batch size $n$.

LoRA parameterizes a rank-$r$ update as $\Delta W = B_\ell A_\ell$ with trainable $A_\ell \in \mathbb{R}^{r \times d_{\text{in}}}$ and $B_\ell \in \mathbb{R}^{d_{\text{out}} \times r}$, so each adapted layer introduces $r(d_{\text{in}} + d_{\text{out}})$ trainable parameters. In PLATE, the update is $\Delta W = BAQ^\top$ with a frozen input basis $Q \in \mathbb{R}^{d_{\text{in}} \times k}$, a trainable core $A \in \mathbb{R}^{r \times k}$, and a frozen output selector $B \in \mathbb{R}^{d_{\text{out}} \times r}$ (stored as a dense matrix in our implementation). This yields $rk$ trainable parameters per layer. For same $r$, when $k \ll d_{\text{in}} + d_{\text{out}}$, PLATE therefore reduces the number of trainable parameters and corresponding optimizer states by a large factor.

In the forward pass, LoRA applies $\Delta W x$ as $(xA^\top)B^\top$, which costs $\mathcal{O}(nd_{\text{in}}r + nrd_{\text{out}})$. In our PLATE implementation, the adapter branch is evaluated as

$$Z := xQ \in \mathbb{R}^{n \times k}, \qquad U := ZA^\top \in \mathbb{R}^{n \times r}, \qquad \Delta y := UB^\top \in \mathbb{R}^{n \times d_{\text{out}}},$$

implemented via dense `torch.nn.functional.linear` calls for both $Q$ and $B$. Consequently, the adapter-branch

forward cost is

$$\mathcal{O}(nd_{\text{in}}k \ + \ nkr \ + \ nrd_{\text{out}}).$$

This aligns with our empirical observation (see Figure 8) that PLATE incurs a modest training-time overhead driven primarily by the additional projection through $Q$

Memory usage has three main components: optimizer states for trainable parameters, frozen adapter buffers, and saved activations needed for backpropagation. LoRA introduces two trainable matrices per adapted layer and therefore requires optimizer states for $r(d_{\text{in}} + d_{\text{out}})$ parameters, whereas PLATE trains only $A \in \mathbb{R}^{r \times k}$ and thus requires optimizer states for only $rk$ parameters. PLATE additionally stores frozen buffers $Q \in \mathbb{R}^{d_{\text{in}} \times k}$ and $B \in \mathbb{R}^{d_{\text{out}} \times r}$ (stored densely in our implementation), which contribute $(d_{\text{in}}k + d_{\text{out}}r)$ in memory but do not create optimizer states.

Crucially, peak training memory is also driven by the activations that autograd must retain to form weight gradients. As a result, LoRA's trainable down-projection requires retaining the full input activation $x \in \mathbb{R}^{n \times d_{\text{in}}}$ (in addition to smaller rank-$r$ intermediates), and this cost accumulates across all adapted layers. In contrast, PLATE keeps both $Q$ and $B$ frozen and trains only $A$, so the adapter branch only needs to retain the projected activation $Z = xQ \in \mathbb{R}^{n \times k}$ to compute $\nabla A$. When $k \ll d_{\text{in}}$, this reduces the adapter-induced activation footprint, and in our DistilBERT experiment the optimizer-state and activation savings dominate the additional frozen-basis storage, yielding lower peak GPU memory for PLATE across hyperparameters (see Figure 8).

## B.5. Computational Analysis for PLATE initialization

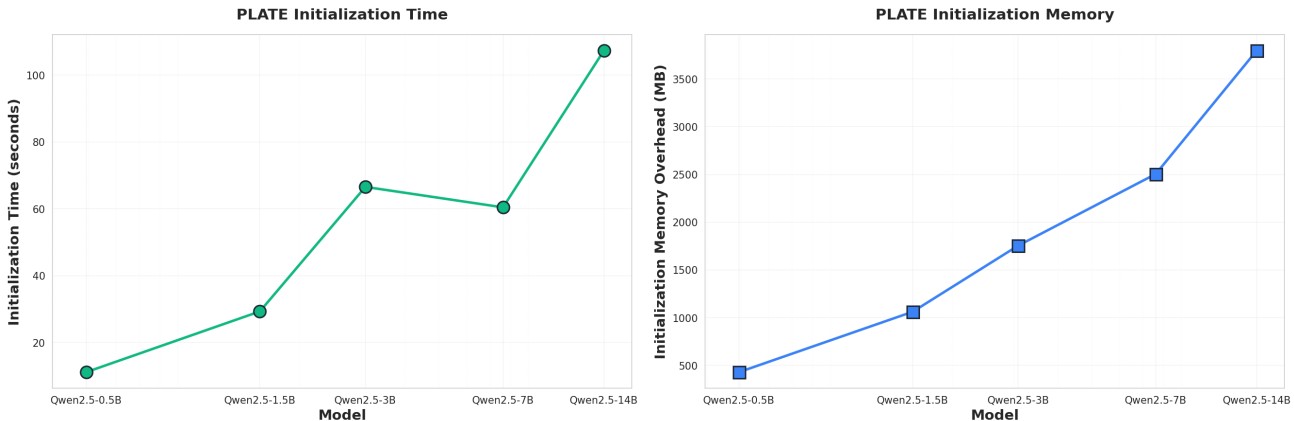

*Figure 9.* **Initialization complexity of PLATE adapters as a function of model size: (left)** Time complexity for computing $Q$ matrices (via SRHT-based eigenproblem solving) and $B$ selection matrices. **(Right)** Peak memory overhead during initialization, which includes both permanent adapter parameters and temporary computation buffers. Experiments conducted on Qwen2.5 models with fixed PLATE hyperparameters ($r = 64, \tau = 0.9$).

## C. Additional Experiments

We now provide additional experiments to Section 5.

### C.1. Synthetic regression with tunable task dissimilarity

To explicitly vary task relatedness and capture how full fine-tuning, LoRA, and PLATE forget/learn when task 1 and task 2 dissimilarity increases. Concretely, we construct two regression tasks over Gaussian inputs $\mathbf{x} \sim \mathcal{N}(0, I_{100})$ with targets $f(\mathbf{x}) = \tanh(\mathbf{w}^{\top}\mathbf{x})$. Task 1 uses a fixed unit vector $\mathbf{w}_1$ while task 2 uses a rotated version of $\mathbf{w}_1$ where $\alpha$ relates to the rotation angle. Task dissimilarity is measured as $D^2(\alpha) = \mathbb{E}[(f_1(\mathbf{x}) - f_{2,\alpha}(\mathbf{x}))^2]$. We use a 2-layer $\tanh$ MLP (512 units). LoRA (rank 8) on all backbone layers, and PLATE ($r = 50, \tau = 0.6$); all methods train 100 epochs per task (Table 3).

Figure 10 shows forgetting and task 2 loss as a function of task dissimilarity, i.e., $D^2(\alpha)$. Full fine-tuning and LoRA exhibit approximately linear growth of forgetting with task dissimilarity; LoRA forgets even more than full fine-tuning despite using only a small subspace. PLATE, in contrast, keeps forgetting an order of magnitude smaller across the entire range, even when the tasks are nearly orthogonal, while having a slightly higher task 2 loss than the other methods. We additionally evaluate the two replay-free baselines. L2-Init is the strongest at preventing forgetting (reaching only $\sim 5 \times 10^{-3}$ MSE drift at maximum task distance, an order of magnitude below Full FT) but at the cost of a higher task-2 loss ($\sim 1.6 \times 10^{-3}$ vs. $\sim 3 \times 10^{-4}$ for LoRA), reflecting the tension between regularization strength and plasticity. O-LoRA closely tracks LoRA and, like LoRA, fails to control forgetting as task dissimilarity grows. PLATE remains the most favorable point on the

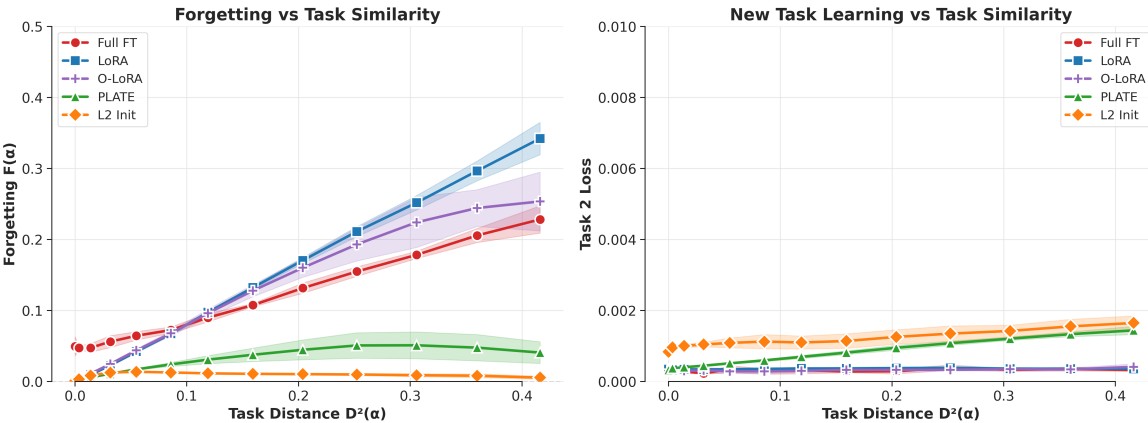

*Figure 10.* **Synthetic regression with tunable task dissimilarity:** (**Left**) forgetting on task 1 (increase in MSE) versus task dissimilarity $D^2(\alpha)$; (**Right**) Task 2 test loss versus $D^2(\alpha)$. Forgetting for full FT (red) and LoRA (blue) grows roughly linearly with $D^2(\alpha)$, while PLATE (green) remains an order of magnitude smaller even for dissimilar tasks, with only a modest increase in Task 2 loss.

curve, matching L2-Init on task-2 loss while keeping forgetting below it for nearly the entire distance range. This synthetic experiment directly supports our theoretical picture: forgetting is governed by the geometry of the allowed update family $S$. In particular, restricting updates to be approximately data-orthogonal and concentrating plasticity on redundant degrees of freedom yields forgetting that grows only weakly with task drift.

### C.2. Worst-case forgetting complementary experiment

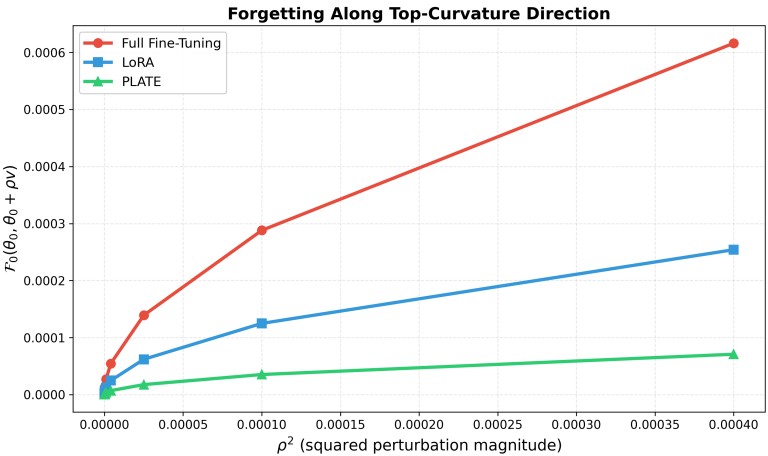

*Figure 11.* **Restricted-curvature forgetting:** We train an MLP on MNIST digits 0-4 to obtain parameters $\theta_0$. For each method, we perturb the trained model by $\theta_0 + \rho v$ and measure the resulting forgetting $\mathcal{F}0(\theta_0, \theta_0 + \rho v) = L_0(\theta_0 + \rho v) - L_0(\theta_0)$ where $v$ is the unit vector in each subspace that maximizes $v^T H_0 v$, i.e., *highest-curvature direction*. PLATE exhibits the smallest slope, indicating substantially reduced restricted curvature and correspondingly smaller worst-case forgetting.

Figure 11 empirically supports the restricted-curvature perspective: even within a parameter-efficient family, the old-task loss can increase rapidly if the family still contains high-curvature directions. In this MNIST experiment, LoRA's low-rank tangent subspace still exposes directions with relatively large restricted curvature, whereas PLATE's structured family exhibits a smaller slope. This is consistent with PLATE's two weight-only restrictions: $(i)$ restricting plasticity to a subset of redundant output channels and $(ii)$ constraining updates to a low-energy input subspace inferred from frozen weights, both designed to reduce drift.

### C.3. Matched-capacity ablations and direct old-model drift

To isolate the contributions of the two PLATE design choices ($B$ and $Q$), we run matched-capacity ablations that keep the adapter architecture and parameter count identical to PLATE but replace the geometric construction of $B$, $Q$, or both,

with the following internal controls. **Random-**$B$ replaces the redundant-neuron selection with a random subset of $r$ output channels; **Least-red.-**$B$ selects the $r$ *least* redundant channels; **Random-**$Q$ replaces the low-energy input basis with a random orthonormal $k$-frame; **Random-**$(B+Q)$ randomizes both; and **High-energy-**$Q$ selects the $k$ *top* (rather than bottom) eigenvectors of $G_{\text{in}}$. All controls share PLATE's parameter count, optimizer, and training schedule.

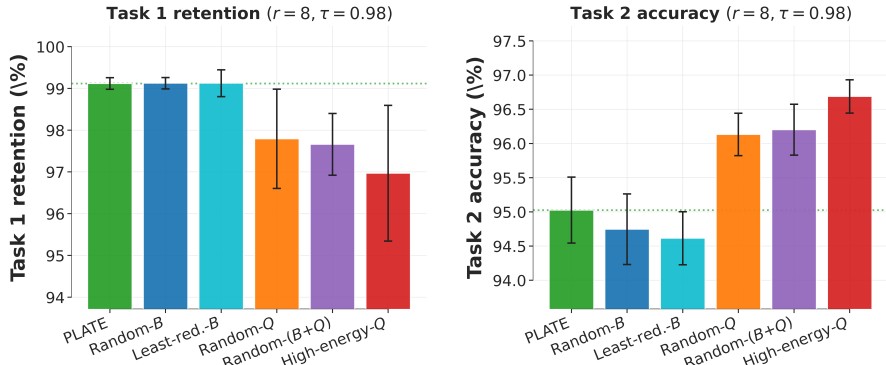

*Figure 12.* **MNIST 0-4→5-9 matched-capacity ablations. (Left)** Task-1 retention at the canonical config ($r=8, \tau=0.98$) averaged over 5 seeds. **(Right)** Task-2 accuracy at the same config. Randomizing the input basis $Q$ (orange, purple, red) consistently lowers Task-1 retention compared to PLATE while slightly boosting Task-2 accuracy, whereas randomizing $B$ (blue, cyan) leaves retention essentially unchanged but slightly hurts new-task adaptation. This empirically separates the two PLATE design choices: $Q$ is the retention driver, $B$ is the adaptation driver.

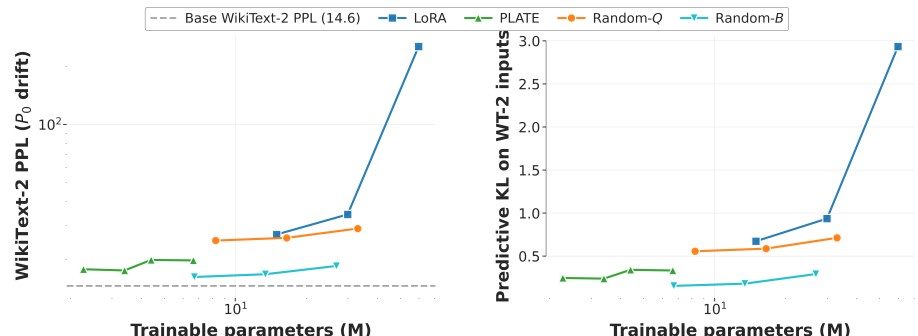

*Figure 13.* **Direct old-model drift on WikiText-2 → Middle English (Qwen2.5-3B). (Left)** WikiText-2 perplexity after task-2 adaptation; the grey dashed line is the base-model PPL. **(Right)** Average predictive KL between the base and adapted models on WikiText-2 inputs. At matched parameter budgets, Random-$Q$ produces $\approx 2-3\times$ larger KL drift than PLATE, while Random-$B$ tracks PLATE closely; LoRA's drift grows sharply with rank and is up to an order of magnitude larger than PLATE's at comparable capacity. This confirms the same $B/Q$ asymmetry observed on MNIST in a language-modeling setting.

These ablations resolve the question of whether the gains come from using *any* restricted update family of the right size or specifically from PLATE's weight-only construction. Replacing $Q$ with a random orthonormal basis at matched parameter count consistently increases both predictive KL on $P_0$ and old-task forgetting, while replacing $B$ with random or anti-redundant selections has little effect on retention but slightly degrades new-task learning. This is consistent with our theoretical framing in Section 3.3: input-side protection via $Q$ controls $\varepsilon(S)$ and thus the worst-case forgetting floor, while output-side restriction via $B$ primarily shapes *where* the plasticity budget is allocated.

### C.4. Weight geometry vs. activation geometry

PLATE constructs the protected input subspace $Q$ from the top-$k$ *right singular vectors of the (frozen) weight matrix* $W$. A natural concern is whether these weight-derived directions actually capture the dominant directions of the empirical activation distribution that the model sees at inference. We test this directly.

For each linear layer with weight $W \in \mathbb{R}^{d_{\text{out}} \times d_{\text{in}}}$ and a set of $N$ input activations $X \in \mathbb{R}^{N \times d_{\text{in}}}$ collected from pretraining-like data (MNIST digits 0–4 for the MLP; WikiText-2 for Qwen2.5-3B), we compare three $k$-dimensional input subspaces: the top-$k$ right singular vectors of $W$ (the PLATE basis, $V_k = [v_1, \ldots, v_k]$); the top-$k$ eigenvectors of $\Sigma_{\text{act}} = \frac{1}{N} X^\top X$ (the

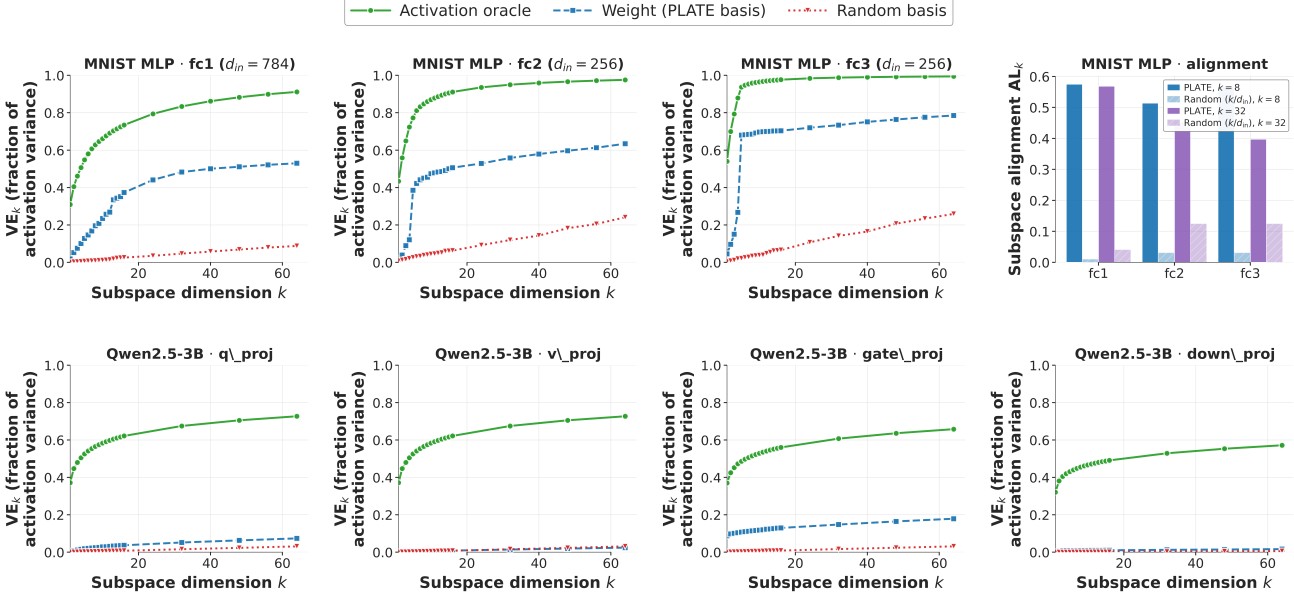

*Figure 14.* **Weight geometry vs. activation geometry.** Activation variance captured by three input subspaces of dimension $k$: the activation oracle (top-$k$ eigenvectors of $\Sigma_{\mathrm{act}}$, green), the weight-derived subspace used by PLATE (top-$k$ right singular vectors of $W$, blue), and a random $k$-frame (red). **(Top)** MNIST MLP layers; **(Bottom)** Qwen2.5-3B averaged across 7 transformer blocks per module type. For most modules, weight-derived subspaces capture substantially more activation variance than random of equal size (e.g., $\sim 4-20\times$ in MNIST MLP layers; $\sim 4-26\times$ in gate_proj, down_proj, q_proj of Qwen2.5-3B), confirming that weight geometry is a useful, data-free proxy for the dominant activation directions. The exception is v_proj, where weight and activation principal directions are nearly orthogonal: this corner case is consistent with the key/value memory view of FFN/attention layers, in which colinearity of one projection's weights need not imply colinearity in the matching activation distribution. The rightmost MNIST panel reports the alignment metric $\mathrm{AL}_k$ for the PLATE weight basis (solid bars) and the chance-level random baseline $k/d_{\mathrm{in}}$ (hatched bars), at $k=8$ (blue) and $k=32$ (purple); PLATE values sit $\sim 5-50\times$ above chance across all layers.

data-driven oracle, $U_k = [u_1, \ldots, u_k]$); and a uniformly random $k$-frame. We report $\mathrm{VE}_k := \frac{\mathrm{tr}(V_k^\top \Sigma_{\mathrm{act}} V_k)}{\mathrm{tr}(\Sigma_{\mathrm{act}})}$, $\mathrm{AL}_k :=$ $\frac{1}{k} \sum_{i=1}^{k} \sum_{j=1}^{k} |\langle u_i, v_j \rangle|^2 = \frac{1}{k} \|U_k^\top V_k\|_F^2$. $\mathrm{VE}_k$ measures the fraction of activation variance captured by the weight-derived subspace; $\mathrm{AL}_k \in [0, 1]$ measures direct geometric overlap between the weight and activation principal directions, and reduces to $\mathrm{AL}_k = 1$ when the two subspaces coincide. For a Haar-random $k$-frame the expected value is $\mathbb{E}[\mathrm{AL}_k] = k/d_{\mathrm{in}}$, which we use as a chance-level reference.

*Table 1.* Variance-explained ratio between the PLATE weight-derived subspace and a random $k$-frame ($\mathrm{VE}_k(W)/\mathrm{VE}_k(\mathrm{random})$). Larger means the weight geometry better matches activation geometry. MNIST values are per layer; Qwen2.5-3B values are averaged across 7 transformer blocks.

| | **MNIST MLP** | | | **Qwen2.5-3B (avg. over 7 blocks)** | | |
|---|---|---|---|---|---|---|
| **Layer** | $d_{\mathrm{in}}$ | $k{=}8$ | $k{=}32$ | **Module** | $k{=}8$ | $k{=}32$ |
| fc1 | 784 | 22.0× | 10.3× | self_attn.q_proj | 6.5× | 3.3× |
| fc2 | 256 | 11.9× | 4.7× | self_attn.v_proj | 1.6× | 0.9× |
| fc3 | 256 | 20.6× | 5.2× | mlp.gate_proj | 26.0× | 8.8× |
| | | | | mlp.down_proj | 11.1× | 4.2× |

This experiment validates a central assumption of PLATE: although $Q$ is built *without ever observing* the activation distribution, it preferentially captures high-variance activation directions much better than chance for most modules. Importantly, $Q$ does *not* match the oracle activation eigenbasis exactly, and for some modules (notably v_proj) the alignment is at the level of chance.

## C.5. Robustness under base-model compression

PLATE's geometric assumption rests on the existence of redundant directions in the pretrained weight matrices. Structured compression methods that remove redundant components from the backbone are therefore a natural stress test for the method.

We use SliceGPT (Ashkboos et al., 2024) to compress Qwen2.5-3B at two pruning ratios (20% and 30%) and apply PLATE on top of each compressed model under the same Middle English adaptation protocol as Section 5.2.1, with $\tau{=}0.98$ and $r \in \{32, 64, 128\}$.

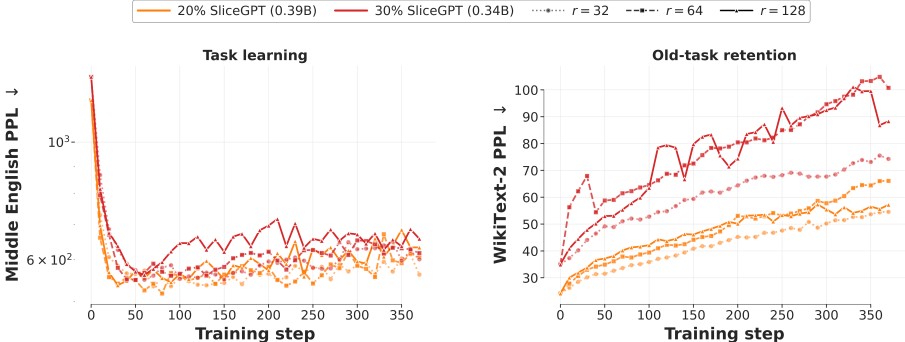

*Figure 15.* **PLATE on SliceGPT-compressed Qwen2.5-3B. (Left)** Middle English perplexity over training (task learning). **(Right)** WikiText-2 perplexity over training (retention). PLATE continues to learn the new task on both compressed backbones (task PPL drops by $\sim 47{-}55\%$ relative to initialization). Retention degrades with the more aggressive compression: WikiText-2 PPL rises more sharply on the 30%-pruned model than on the 20%-pruned one, consistent with the fact that pruning removes some of the redundant directions PLATE relies on. The behaviour is continuous rather than abrupt.

These results confirm that PLATE's mechanism degrades smoothly under compression: adaptation continues to work on both compressed backbones (task PPL still drops substantially), while retention becomes harder as the redundancy budget PLATE exploits is more aggressively removed by pruning. We view this as evidence that PLATE remains usable on compressed deployments, with the expected caveat that stronger compression reduces the headroom of the retention–plasticity trade-off.

### C.6. Behaviour on longer task sequences ($T \geq 3$)

Our main experiments focus on the canonical two-task continual-learning setup. For completeness, we evaluate PLATE under a sequential protocol with four tasks, using the recursive construction described in Appendix B.3. Following the original setup of Section 5.2.2, we train a 3-layer ReLU MLP on a sequence of MNIST binary tasks (2v3 $\rightarrow$ 4v5 $\rightarrow$ 6v7 $\rightarrow$ 8v9) and report all 20 $(r, \tau)$ configurations from the main sweep.

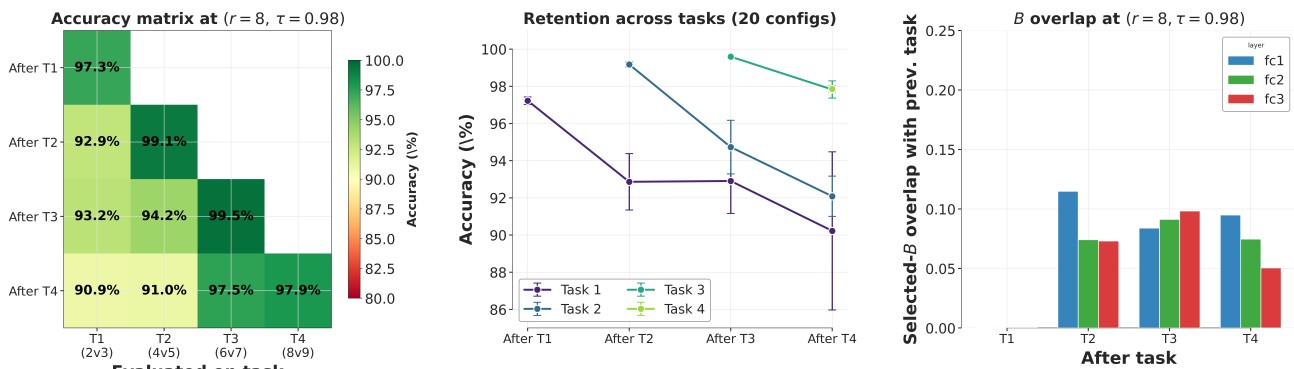

*Figure 16.* **PLATE under the four-task sequential MNIST protocol. (Left)** Accuracy matrix at the canonical $(r{=}8, \tau{=}0.98)$ config: rows are training stages, columns are evaluation tasks. **(Middle)** Mean accuracy on each previously seen task across all 20 sweep configurations, as a function of how many tasks have been trained. **(Right)** Selected-$B$ overlap with the previous task at the canonical config: small overlaps indicate that successive tasks pick mostly disjoint trainable channels from PLATE's redundant pool.

Two observations support that PLATE behaves sensibly beyond two tasks. First, the diagonal of the accuracy matrix stays at $\geq 97\%$ on every newly trained task: new-task adaptation is preserved across the sequence. Second, the off-diagonal entries decay progressively: at the canonical $(r{=}8, \tau{=}0.98)$ config, accuracy on Task 1 drops from 97.3% at the end of Task 1 to 90.9% at the end of Task 4, and similar mild decay holds for intermediate tasks. The aggregate retention curve confirms this trend across the entire sweep, with the predictable $(r, \tau)$ dependence: more conservative configurations (smaller $r$, larger $\tau$) accumulate less forgetting over the stream.

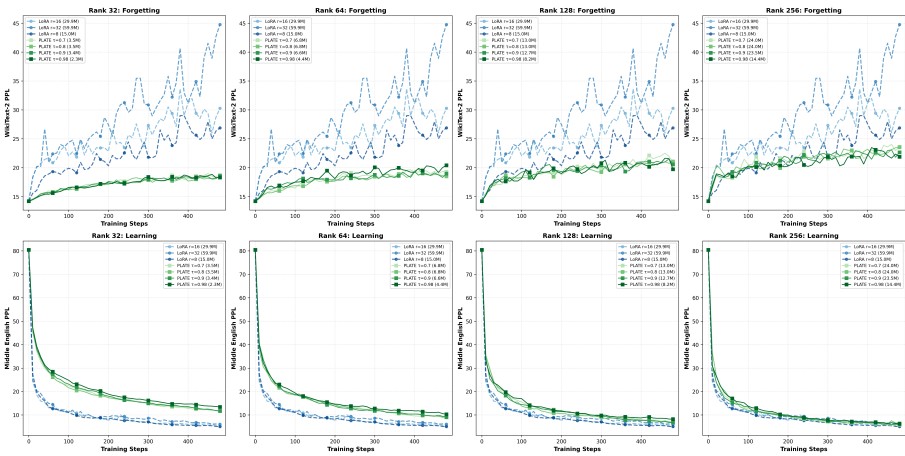

*Figure 17.* **Qwen 2.5-3B - Extended PLATE sweep across** $(r, \tau)$**:** Columns fix the PLATE output rank $r \in \{32, 64, 128, 256\}$ and sweep $\tau \in \{0.70, 0.80, 0.90, 0.98\}$ (green, solid) against LoRA baselines with varying ranks (blue, dashed). Top row reports WikiText-2 perplexity (forgetting) and bottom row reports Middle English perplexity (task learning), both over training steps.

## C.7. Additional parameter sweep for Middle-English experiment & binary classification
## D. Experimental Details

*Table 2.* Hyperparameter grids and target modules for each adaptation method.

| Method | Hyperparameter | Values |
|---|---|---|
| **LoRA** | Rank $r$ | Vision/Regression/NLP: $\{1, 8, 16, 32, 64, 128\}$; LLM: $\{8, 16, 32\}$ |
| | Scaling $\alpha$ | $\alpha/r = 0.5$ |
| | Target modules | Vision/Regression/NLP: all linear layers; LLM: attention and MLP projections (e.g., `q, k, v, o, up, down, gate`). |
| **PLATE** | Rank $r$ | Vision: $\{32, 64, 128, 256, 350\}$; NLP: 32; Regression: 50; LLM: $\{32, 64, 128, 256\}$ |
| | Energy threshold $\tau$ | Vision/NLP: $\{0.6, 0.7, 0.8, 0.9\}$; LLM: $\{0.70, 0.80, 0.90, 0.98\}$. |
| | Max rank $r_{\max}$ | $\{256, 512\}$ |
| | Scaling $\rho$ | 0.5 |
| | Target modules | Vision/Regression/NLP: all linear layers. |
| **Full FT** | Trainable parameters | All parameters. |

*Table 3.* Domain-specific experimental configurations. All experiments follow Algorithm 3 with $K = 10$ runs.

| Domain | Tasks $\mathcal{D}_1 \rightarrow \mathcal{D}_2$ | Architecture | Epochs $(E_1/E_2)$ | Learning Rate | Batch Size |
|---|---|---|---|---|---|
| **Vision** | MNIST 0-4 $\rightarrow$ 5-9 (5 classes each) | 3-layer ReLU MLP (784$\rightarrow$256$\rightarrow$256$\rightarrow$256) + two 256$\rightarrow$5 heads | 10/10 | $10^{-3}$ (Adam) | 128 |
| **NLP (small)** | AG News $\rightarrow$ IMDB (4 classes $\rightarrow$ 2 classes) | DistilBERT-base (6 layers, 768-dim) + two linear heads (768$\rightarrow$4/2) | 3/3 | $2 \times 10^{-5}$ (AdamW, 10% warmup) | 32 |
| **Regression** | $f_1(\mathbf{x}) \rightarrow f_2^\alpha(\mathbf{x})$ on Gaussian inputs ($d = 100$) | 2-layer tanh MLP (100$\rightarrow$512$\rightarrow$512) + two 512$\rightarrow$1 heads | 100/100 | $10^{-3}$ (Adam) | 128 |
| **LLM** | WikiText-2 $\rightarrow$ Middle English (EN-ME) | Qwen/Qwen2.5-3B (Causal LM), adapters on attention/MLP projections | 0/1 | $10^{-3}$ (AdamW) | 16 |

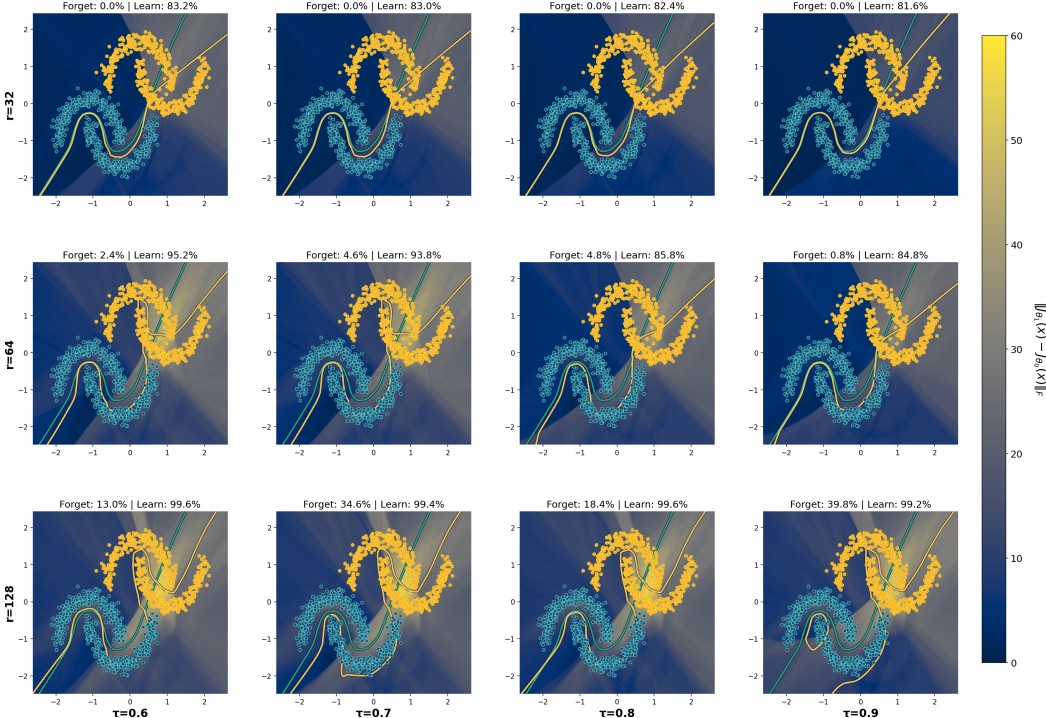

*Figure 18.* **Local-geometry view of forgetting - Extended PLATE sweep across** $(r, \tau)$**:** We sweep PLATE's $(r, \tau)$ on a two-moons continual-learning toy: increasing $r$ expands the plasticity budget and improves task 2 performance but can increase task 1 drift/forgetting, while increasing $\tau$ tends to concentrate updates onto more redundant degrees of freedom and reduces drift/forgetting. Overall, PLATE provides an explicit mechanism to target a desired point on the retention-adaptation trade-off.

---

**Algorithm 3** Parameter-efficient two-task continual learning protocol

---

**Require:** $\mathcal{D}_1, \mathcal{D}_2$, base model $f_\theta$, heads $h_1, h_2$, methods $\mathcal{M} = \{\text{Full-FT}, \text{LoRA}, \text{PLATE}\}$, configs $\mathcal{C}_m$
  1: **for** $k = 1$ to $K$ **do**
  2:     Initialize $f_\theta^{(k)}, h_1^{(k)}, h_2^{(k)}$
  3:     **Stage 1:** train $(f_\theta^{(k)}, h_1^{(k)})$ on $\mathcal{D}_1$ for $E_1$ epochs
  4:     Record baseline Task 1 accuracy and save checkpoint $\theta_1^{(k)}$
  5:     **for** method $m \in \mathcal{M}$ **do**
  6:       **for** config $c \in \mathcal{C}_m$ **do**
  7:         Restore $f_{\theta^{(k)}} \leftarrow f_{\theta_1^{(k)}}$, freeze $h_1^{(k)}$
  8:         Apply adapters for $(m, c)$ to $f_{\theta^{(k)}}$ and define trainable set $\Theta$ (plus $h_2^{(k)}$)
  9:         **Stage 2:** train $(f_{\theta^{(k)}}, h_2^{(k)})$ on $\mathcal{D}_2$ for $E_2$ epochs
10:         Evaluate Task 2 (learnability) & Task 1 (forgetting)
11:       **end for**
12:     **end for**
13: **end for**
14: Aggregate means and standard deviations across $k$ for each $(m, c)$

---

# E. Scope and Limitations

Our framing of PLATE makes a number of assumptions that are worth stating explicitly. $(i)$ The bound in Proposition 3.2 uses a global smoothness constant $\beta$ such that $\nabla_f^2 \ell(f_{\theta_0}(x), y) \preceq \beta I$. This is satisfied by the losses used in the paper (e.g., binary logistic with $\beta = \frac{1}{4}$, multiclass cross-entropy with $\beta = \frac{1}{2}$), but the relevant local curvature can be smaller at confident operating points and the bound is therefore conservative away from decision boundaries. $(ii)$ Proposition 3.2 drops a residual second-order term in the Hessian decomposition for analytical clarity. Empirically (Appendix A.6) this term is not exactly

zero; the Gauss–Newton lens becomes most informative inside the PLATE subspace and less so on generic backbone or LoRA directions. $(iii)$ PLATE uses weight-space similarity as a heuristic, data-free *proxy* for functional redundancy. It is not a claim of exact functional equivalence: in the piecewise-affine view of MLPs, weight colinearity reduces the chance of new input partitions but does not control the output side of those partitions, and the alignment with activation geometry varies across modules (Appendix C.4). $(iv)$ PLATE's mechanism relies on the existence of redundant directions in the pretrained backbone. Aggressive structured compression removes some of this redundancy; the method degrades smoothly but with a worse retention–plasticity trade-off (Appendix C.5). $(v)$ Most of our experiments target the two-task setting that is the focus of the theory. We provide a recursive multi-task protocol (Appendix B.3) and empirical evidence on a four-task MNIST stream (Appendix C.6); the claims of this paper are best read as targeting data-free domain adaptation and short continual-learning streams rather than open-ended lifelong learning.

