# OpenReview forum: "PLATE: Plasticity-Tunable Efficient Adapters for Geometry-Aware Continual Learning"
_ICML.cc/2026/Conference — ICML 2026 regular_

### Official Review · Reviewer_vGde · 2026-03-03

**Soundness:** 3
**Presentation:** 4
**Significance:** 3
**Originality:** 3
**Overall Recommendation:** 5
**Confidence:** 4

**Summary:**

This paper proposes a parameter-efficient continual adaptation method for pretrained models in a setting where old-task data are unavailable. For each linear layer, PLATE parameterizes the update as
$\Delta W = B A Q^\top$,

where \(B\) selects a subset of “redundant” output channels (trainable neurons), \(Q\) spans a low-energy input subspace computed from the frozen weights, and only \(A\) is trained on the new-task data. This yields an explicit, interpretable retention–plasticity trade-off controlled by two knobs: \(r\) (the number of trainable output channels) and ($\tau$) (an energy threshold that determines how strongly updates are restricted to low-energy directions).

The method is motivated through a local-geometry view of forgetting based on Jacobian drift, and the theory connects worst-case forgetting under the restricted update family to a **restricted curvature** quantity, which is then related to functional drift. Empirically, PLATE is evaluated mainly against full fine-tuning and LoRA across small vision/NLP settings and an LLM adaptation experiment.

**Compliance With Llm Reviewing Policy:**

Affirmed.

**Final Justification:**

The paper proposes a data-free, parameter-efficient continual adaptation method that uses geometric redundancy in pretrained weights to build structured low-rank updates with an explicit retention–plasticity tradeoff. My initial concerns were the limited evaluation scope, the theory-to-algorithm gap, and the validation of the redundancy assumption.

The rebuttal addressed these concerns convincingly and thus I change my evaluation from 4 to 5.

**Key Questions For Authors:**

- Residual curvature term (Prop. 3.2): In the Gauss–Newton + residual decomposition, the residual is assumed bounded and set to 0 “for clarity.” Do you have empirical evidence (for your losses/models) that this residual is small along PLATE update directions relative to the Gauss–Newton term?

- Bounded $\nabla_f^2 \ell$ assumption: Under which concrete loss/model regimes do you assume $\nabla_f^2 \ell \preceq \beta I$? In your experiments, is $\beta$ finite/tight, and how does it depend on logit magnitude/calibration or the local operating point?
-  Multi-task saturation behavior: How does PLATE behave on longer streams  and how does performance degrade as tasks accumulate? In particular, does the “redundant” pool shrink over time, and does the overlap in selected B channels increase?
- Mechanism diagnostics: Can you report simple layer-wise diagnostics across tasks such distribution of redundancy scores used to pick B, or overlap of selected channels across tasks ?

**Limitations:**

The paper would benefit from a more explicit limitations discussion, particularly around the likely failure modes over long task streams, the extent to which the theoretical story depends on the residual-curvature idealization  and an empirical clarification of what “redundant/dormant” channels mean in practice and when this assumption may break down.

**Strengths And Weaknesses:**

Strengths:

- Clear, reproducible algorithm with weight-only preprocessing. PLATE is precisely specified through the adapter family
$\Delta W = B A Q^\top$,
with a clean separation between one-time computation of (B, Q) from frozen weights and training only A. This design makes the method easy to implement, evaluate, and reproduce.
- Interpretability and explicit control of the stability–plasticity trade-off. The roles of r (output-channel plasticity budget) and \tau (conservatism via energy thresholding / subspace restriction) are clearly articulated. The paper also includes knob sweeps (toy + LLM setting) that demonstrate how these parameters affect adaptation vs. retention.
- Plausible and Solid theoretical rationale. The analysis leverages local quadratic expansions and Jacobian/curvature  decompositions to relate restricted updates to functional drift, which is consistent with common tools used to reason about forgetting and stability in neural networks.

Weaknesses:
- Redundant channel empirical validation. PLATE relies on selecting “redundant” output channels via a weight-similarity criterion. The case would be stronger with direct evidence that these channels are redundant in function space (not just similar in weights), and with an analysis of whether this redundancy pool persists or depletes as tasks accumulate.
- Long-horizon continual behavior remains unclear. Much of the intuition and several key visualizations are framed around a two-task $P_0 \rightarrow P_1$ regime. Continual learning typically could potentially involve longer task streams where interference accumulation and capacity saturation become central. The current evidence does not clearly characterize whether PLATE’s redundancy/plasticity budget saturates over longer horizons, or how performance evolves as the number of tasks grows.

---

> ### Author Rebuttal · Authors · 2026-03-30
>
> **1.Residual curvature term (Prop. 3.2)**
>
> Thank you, this is a very useful question. We agree our original wording was too strong, and we will soften it in the revision.
>
> We ran a controlled MNIST diagnostic at the old-task solution $\theta_0$ on old-task data. For directions $v$ in the update family, we measured the directional old-task curvature
> $v^\top H_0 v,$
> and decomposed it into its Gauss–Newton contribution and residual contribution, following the same Hessian decomposition used in the proof discussion of Prop. 3.2.
>
> Our main finding is that the residual term is not negligible in absolute magnitude, including along PLATE directions. So we do not claim that the old-task Hessian is well approximated by its Gauss–Newton part alone along PLATE directions.
>
> However, the more important takeaway is that the Gauss–Newton contribution is much more pronounced along PLATE directions than along the comparison directions. After matching parameter count and scale, the median fraction of curvature explained by the Gauss–Newton term is about $3.3\%$ for PLATE, versus about $0.2\%$ for both generic backbone directions and LoRA directions.
> Thus PLATE does not make the residual disappear, but it is the only tested update family in which the Gauss–Newton / drift component becomes materially visible. In other words, among the tested subspaces, PLATE is the one for which the drift-based geometric lens underlying Prop. 3.2 is the most informative empirically. This is consistent with the motivation of PLATE: it is not an arbitrary restriction of the update space, but a structured subspace chosen to better align adaptation with old-task geometry.
>
> We will revise the paper to make this distinction explicit: the residual-free approximation is not exact in our controlled setting, but the Gauss–Newton / functional-drift component is substantially more relevant inside the PLATE subspace than in generic or LoRA directions.
>
>
> **2.Bound assumption**
>
> Thank you, this is an important clarification. In Proposition 3.2, the assumption
>
> $$
> \nabla_f^2 \ell(f_{\theta_0}(x), y) \preceq \beta I
> $$
>
> is a standard bounded-curvature assumption on the loss with respect to the model outputs/logits $f$, and it is satisfied for the losses used in our experiments. In particular, for binary logistic loss with logit $f$,
>
> $$
> \frac{d^2 \ell}{d f^2} = \sigma(f)\bigl(1-\sigma(f)\bigr) \le \frac{1}{4},
> $$
>
> so one may take $\beta = \frac{1}{4}$ globally. For multiclass cross-entropy, one may take the global bound $\beta = \frac{1}{2}$. Therefore, in the classification and language-modeling settings studied in the paper, $\beta$ is finite, so the assumption is not vacuous. At the same time, we agree that the relevant quantity in practice is the local curvature, which depends on the operating point through the predictive distribution: it is largest near uncertain examples and typically smaller for highly confident predictions. We will revise the text around Proposition 3.2 to make these concrete instantiations explicit and to clarify that $\beta$ is used as a local smoothness constant, not as an empirically tuned parameter.
>
>
>
> **3.Multi-task saturation behavior**
>
> Protocol. We evaluate PLATE under a sequential continual learning protocol:
> $W_0$  ←  pretrain on task 0
> for t = 1, 2, ..., T:
>    $Q_{in}$  ←  bottom-k eigenvectors of $W_{t-1}^T W_{t-1}$   # redundant pool
>    A, B  ←  randomly initialised rank-r adapter in Q_{in}
>    train (A, B) on task t,  $W_{t-1}$ frozen
>    $W_t$   ←  $W_{t-1}  +  \rho · B A Q_{in}^T$              # merge
>    save head_t
>
> Setup. 3-layer ReLU MLP (256 hidden, pretrained on MNIST 0v1). Sequential tasks: 2v3 → 4v5 → 6v7 → 8v9.
>
> Accuracy (%):
> | After \ Eval | T1 (2v3) | T2 (4v5) | T3 (6v7) | T4 (8v9) |
> | ------------ | -------: | -------: | -------: | -------: |
> | After T1     |     97.0 |        — |        — |        — |
> | After T2     |     93.4 |     99.2 |        — |        — |
> | After T3     |     93.8 |     95.1 |     99.5 |        — |
> | After T4     |     92.3 |     93.2 |     98.0 |     97.9 |
>
> Redundant neuron pool (effective rank of $W W^T$):
>
> |                 |   fc1 |   fc2 |   fc3 |
> | --------------- | ----: | ----: | ----: |
> | Pretrained init | 161.3 | 144.7 | 146.3 |
> | After T1        | 134.2 | 143.7 | 145.9 |
> | After T2        | 117.9 | 139.3 | 142.4 |
> | After T3        | 109.1 | 138.8 | 141.8 |
> | After T4        | 104.4 | 137.8 | 141.8 |
>
> The redundant neuron pool decreases monotonically with each task confirming the reviewer's intuition that merged updates concentrate the weight spectrum. Critically, learnability remains above 97% throughout and retention stays above 92%, demonstrating that PLATE can sustain strong performance across 4 sequential tasks even as the pool shrinks. The graceful degradation of effective rank suggests the pool is consumed gradually and proportionally to layer capacity.
>
> **4. Mechanism diagnostics**
> Not enough characters to report here. Will add to the paper. Thanks for suggesting

---

> > ### Author Rebuttal · Reviewer_vGde · 2026-04-03
> >
> > I appreciate the authors’ detailed rebuttal. The additional experiments across their responses to multiple reviewers strengthen the empirical case for the work. In light of this, I have decided to increase my score.

---

> > > ### Author Response · Authors · 2026-04-03
> > >
> > > We really appreciate your feedback and the score increase. We noticed the portal is still showing the original score, we just wanted to send a quick note in case the update didn't go through!

---

### Official Review · Reviewer_bu57 · 2026-03-06

**Soundness:** 2
**Presentation:** 3
**Significance:** 3
**Originality:** 3
**Overall Recommendation:** 3
**Confidence:** 4

**Summary:**

This paper introduces PLATE (Plasticity-Tunable Efficient Adapters), a data-free continual learning (CL) framework designed for pretrained foundation models. The core problem addressed is catastrophic forgetting during sequential adaptation when old-task data is strictly unavailable. The authors' central contribution is leveraging the "geometric redundancy" of pretrained networks to construct approximately protected update subspaces directly from the pretrained weights. By restricting adapter updates (plasticity) to these redundant directions, the method aims to learn new tasks without interfering with the dominant feature directions of the pretraining era.  In the landscape of Parameter-Efficient Fine-Tuning (PEFT) and Continual Learning, this work attempts to bridge the gap by shifting the dependency from data-driven activation statistics to weight-driven geometric properties, which is highly desirable for privacy-preserving foundation model adaptation.

**Compliance With Llm Reviewing Policy:**

Affirmed.

**Key Questions For Authors:**

Recursive Definition for $T \ge 3$: Please explicitly define the mathematical protocol for the third task and beyond. Do you freeze $\theta_1$ and compute a new $B_2$ and $Q_2$? If so, how do you mathematically guarantee that updates orthogonal to the $\theta_1$-derived subspace do not interfere with the original $\theta_0$ pretraining knowledge? If you reuse $B_1$ and $Q_1$, how do you prevent the rank-restricted matrix $A$ from saturating and halting learning?

Weight Geometry vs. Activation Geometry: You construct protected subspaces directly from static pretrained weights. Please provide a rigorous empirical measurement showing how tightly the singular vectors of the static weight matrices correlate with the actual empirical activation covariance of the pretraining data.

Sensitivity to Base Model Compression: If the base model $f_\theta$ undergoes structured pruning, the "redundant neurons" utilized by your method will be drastically reduced. Please provide an ablation study demonstrating PLATE's performance on a compressed base model to test the limits of your geometric assumption.

**Limitations:**

The authors need to be far more transparent about the capacity limits of their protected subspaces and the architectural constraints of their base models. The authors must add a dedicated Limitations section addressing two specific dimensions:

Capacity Saturation: The authors must explicitly discuss what happens when the "redundant" subspace is fully saturated after multiple tasks. Does the model gracefully degrade, or does catastrophic forgetting suddenly spike?

Evaluation Scope: The authors should transparently acknowledge that evaluating strictly on a "two-task protocol" does not fully capture the dynamics of lifelong learning, and the claims should be scoped appropriately to "data-free domain adaptation" rather than open-ended continual learning.

**Strengths And Weaknesses:**

Strengths:

Originality: Extracting protected feature directions directly from the redundant neurons of pretrained weights, rather than relying on episodic memory or activation distributions, introduces a refreshing perspective. It cleverly repurposes the inherent over-parameterization of modern foundation models as a geometric shield against forgetting.

Quality: The formulation of routing plasticity into specific subspaces is mathematically elegant and naturally fits the privacy constraints of real-world foundation model fine-tuning.

Significance: Data-free continual learning is a critical bottleneck for deploying foundation models across dynamic, proprietary domains. If scalable, this purely weight-driven geometric approach could significantly reduce the system overhead of traditional CL frameworks.

Weaknesses:

Soundness Issues: While the method is packaged as a general "Continual Learning" framework, it severely lacks a recursive mathematical definition and capacity analysis for sequences where $T \ge 3$. Algorithm 1 only defines the update $\Delta W = BAQ^\top$ relative to a single frozen weight matrix $W^{(l)}$ (which effectively equates to a $T=2$ single downstream fine-tuning protocol). This creates a theoretical dilemma for a third task or beyond: if one recomputes the bases $B$ and $Q$ based on the newly adapted weights, the original protected subspace derived from the pretraining data will suffer from severe recursive drift; if one strictly retains the initial $B$ and $Q$ derived from $\theta_0$, the available capacity restricted by the initial geometric redundancy (the dimension of $A$) will rapidly approach catastrophic exhaustion. The lack of a formal mechanism and empirical validation for handling $T \ge 3$ fundamentally undermines its soundness as a "continual" algorithm.

Novelty Scrutiny: Null-space projection and orthogonal gradient updates are well-explored paradigms in continual learning (e.g., OWM, Adam-NSCL). The primary delta here is shifting the subspace origin from data-driven to weight-driven. However, whether static, linear weight geometry can accurately proxy the non-linear activation manifolds crucial for task performance lacks rigorous mathematical and empirical justification.

Generalization: The method relies heavily on the premise that the base model possesses "substantial geometric redundancy". In industrial practice, base models are often heavily optimized via structured pruning or quantization, largely squeezing out such redundancy. The paper fails to provide evidence that this method remains robust on compressed models.

---

> ### Author Rebuttal · Authors · 2026-03-30
>
> **1. Please explicitly define the mathematical protocol for the third task and beyond**
> Here is a shorter version:
>
> For (T>2), after task (t) we merge the adapter into the backbone and recompute PLATE on the updated weights. Concretely, for task t we freeze $W_{t-1}$, compute $(B_t,Q_t)$ from $W_{t-1}$, train only $A_t$ with $\Delta W_t=\rho B_tA_tQ_t^T$, and then set $W_t=W_{t-1}+\Delta W_t$. Thus $Q_{t+1}$ is computed from the merged backbone $W_t$, not from the original pretrained model, so the admissible update subspace is refreshed after each task. We do not claim this guarantees monotonic expansion or zero interference; the point is that PLATE is not tied to one fixed Q-subspace, and its plastic/protected decomposition adapts to the evolving model geometry.
>
> **2. Weight Geo vs. Activation Geo**
> A figure will be added to the paper, here we summarize the experiment we did run and put some of the results into tables to answer this important question.
>
> Setup. For each pretrained weight matrix W and a set of N input activations X collected from pretraining-like data, we compute two objects:
>
> $V_k$:  top-k right singular vectors of W. These induce PLATE's orthogonal subspace.
> $U_k$:  top-k eigenvectors of the empirical activation covariance Σ_act = (1/N) X^T X.
>
> We report two complementary metrics for each layer at each k:
>
>
> | Metric                      | Formula                                         | Interpretation                                                          |
> | --------------------------- | ----------------------------------------------- | ----------------------------------------------------------------------- |
> | $VE_k$ (variance explained) | $tr(V_k^T \Sigma_{act} V_k) / tr(\Sigma_{act})$ | Fraction of activation variance captured by the weight-derived subspace |
> | $AL_k$ (subspace alignment) | $(1/k),|U_k^T V_k|_F^2$                         | Mean squared cosine of principal angles between the two subspaces.      |
>
> Results: MNIST MLP (3-layer ReLU, trained on digits 0–4):
>
> | Layer       |  k | VE_k (W) | VE_k (Oracle) | VE_k (Random) |  AL_k | AL_k (random baseline) |
> | ----------- | -: | -------: | ------------: | ------------: | ----: | ---------------------: |
> | fc1 (d=784) |  8 |    19.5% |         62.9% |         0.89% | 0.366 |                  0.010 |
> | fc1         | 32 |    48.3% |         83.4% |         4.68% | 0.405 |                  0.041 |
> | fc2 (d=256) |  8 |    45.0% |         84.7% |         3.77% | 0.329 |                  0.031 |
> | fc2         | 32 |    55.8% |         95.0% |        11.94% | 0.289 |                  0.125 |
> | fc3 (d=256) |  8 |    68.6% |         95.9% |         3.33% | 0.432 |                  0.031 |
> | fc3         | 32 |    73.4% |         98.8% |        14.14% | 0.234 |                  0.125 |
>
>
> At k=32: AL_k is 5–10× above the random baseline across all layers. VE_k is 4.7–10.3× above random.
>
> Results: Qwen2.5-3B base model MLP layers (X = WikiText-2 activations):
>
> | Layer               | VE_k (W) | VE_k (Random) | W/Rand |  AL_k | AL_rand | AL/rand |
> | ------------------- | -------: | ------------: | -----: | ----: | ------: | ------: |
> | gate_proj (d=2048)  |    13.0% |         1.70% |   7.6× | 0.093 |   0.016 |    5.9× |
> | up_proj (d=2048)    |     5.0% |         1.70% |   2.9× | 0.066 |   0.016 |    4.2× |
> | down_proj (d=11008) |     0.5% |         0.31% |   1.6× | 0.006 |   0.003 |    2.2× |
>
> The story is consistent with MNIST: gate_proj and up_proj show $AL_k$ 4–6× above the random baseline at k=32 (and up to 20× at k=8). down_proj again is the weak case (~2×).
>
>
>
>
>
> **3.Generalization**
> (Figure will be added to the paper showing the PPL learning/forgetting evolution throughout training) Here is the takeaway.
>
> | Configuration                | Init orig PPL | Final orig PPL | Forget% | Init task PPL | Final task PPL | Task learned |
> | ---------------------------- | ------------: | -------------: | ------: | ------------: | -------------: | ------------ |
> | No compression (0.49B), r=32 |          16.5 |           26.2 |    +59% |          1246 |            384 | ✓ 69% drop   |
> | 20% SliceGPT (0.39B), r=32   |          24.2 |           54.6 |   +126% |          1199 |            563 | ✓ 53% drop   |
> | 30% SliceGPT (0.34B), r=32   |          35.0 |           74.3 |   +112% |          1328 |            602 | ✓ 55% drop   |
>
> PLATE remains effective after SliceGPT compression: task PPL still drops by 53–55%, versus 69% without compression. At the same time, forgetting increases as compression removes redundant directions: original-task PPL rises from +59% in the uncompressed model to +112–126% under 20–30% compression. This is consistent with our hypothesis: PLATE benefits from redundant “safe” directions for adaptation, and when compression removes part of that redundancy, the method still works but with a worse retention–plasticity trade-off. Overall, the geometric assumption degrades gracefully rather than failing abruptly.

---

> > ### Author Rebuttal · Reviewer_bu57 · 2026-04-02
> >
> > I thank the authors for their feedback. I have no further questions.

---

> > > ### Author Response · Authors · 2026-04-03
> > >
> > > Thank you for noting that your concerns are fully resolved. Since the portal hasn't updated, we wanted to check if you still intended to adjust the score to help support the paper in the final decision.

---

### Official Review · Reviewer_qzU7 · 2026-03-09

**Soundness:** 3
**Presentation:** 3
**Significance:** 3
**Originality:** 3
**Overall Recommendation:** 4
**Confidence:** 3

**Summary:**

This paper studies continual adaptation of pretrained models under the realistic constraint that the original pretraining data are unavailable. The main proposal is a parameter-efficient update family of the form $\Delta W = BAQ^\top$, where $B$ selects a subset of output neurons, $Q$ is a frozen low-dimensional input basis constructed from pretrained weights, and only the middle matrix $A$ is trained. The motivation is to reduce forgetting by restricting updates to directions that are expected to induce small drift on the pretrained model's behavior. The paper provides a theoretical discussion connecting old-task forgetting to function drift / restricted curvature, and uses this as motivation for the PLATE design. Empirically, the method shows promising retention-plasticity trade-offs and appears to outperform LoRA in several settings.

**Compliance With Llm Reviewing Policy:**

Affirmed.

**Final Justification:**

The paper presents an interesting and practically relevant approach to data-free continual adaptation through a structured PEFT update family. Its main strengths are the clear problem formulation, the originality of using pretrained weight geometry to construct restricted update subspaces, and promising empirical results showing a favorable retention–plasticity trade-off relative to LoRA. The theory does not fully derive the algorithm, but it provides a useful lens by relating forgetting to drift.

My main concern in the original review was that the specific constructions of $B$ and $Q$ seemed under-validated, with an important gap between the geometric story and the actual empirical evidence. The rebuttal addressed this concern substantially. In particular, the added matched-capacity controls with random $B$, random $Q$, and random $(B+Q)$, together with the direct drift measurements, make the mechanism much better supported. I also appreciate that the authors clarified that weight-space similarity is only a heuristic proxy for functional redundancy, rather than a claim of exact functional equivalence.

The rebuttal addressed the core issue that determined my initial score, and overall I now view the paper as technically solid with meaningful potential impact. I therefore moved from weak reject to weak accept.

**Key Questions For Authors:**

1. **Can the authors provide matched-capacity control ablations with random $B$, random frozen orthogonal $Q$, and random $B$+$Q$?**
   This is the single most important missing analysis for me. These comparisons would help distinguish whether the gains come from the specific geometry-aware construction, or simply from using a more restricted update family. Maybe also consider  an ablation comparing the proposed neuron selection rule for $B$ against alternative choices, such as random neuron selection or selecting the least similar / least redundant neurons. A strong result here would significantly improve my evaluation.

2. **Is weight-space similarity meant as a proxy for functional redundancy, or is the claim stronger?**
   As currently written, the discussion may overstate the connection between parameter-space geometry and functional interchangeability. A clearer and more careful framing here would improve the paper, especially given the FFN key-value-memory perspective.

3. **Can the authors provide more direct evidence or argument that the proposed $B,Q$ construction indeed reduces old-model drift relative to the random controls above (for example, via hidden-state drift, output drift, or logit KL on old-distribution proxies)?**
   Since the theory centers on small drift, this would create a much tighter connection between the theoretical motivation and the empirical mechanism. A convincing result would positively affect my evaluation.

**Limitations:**

yes

**Strengths And Weaknesses:**

## Strengths

- The paper studies an important and realistic problem setting: continual adaptation without access to the original pretraining data, and proposed model-dependent update parameterization $\Delta W = BAQ^\top$.
- The theory section formalizes the connection between output drift and forgetting. While the result is not especially surprising, it still provides a useful lens for thinking about retention-oriented PEFT design.

- The empirical results are promising and suggest that PLATE may achieve a better retention-plasticity trade-off than LoRA in several settings.


## Weaknesses

- The constructions of $B$ and $Q$ appear largely heuristic. The paper gives an intuitive story for selecting redundant output neurons and low-energy input directions, but the justification for these specific choices is not yet sufficiently convincing.
- There is a noticeable gap between the theory and the algorithmic design. The theory motivates the importance of low-drift update families, but it does not directly derive the specific instantiation of $B$ and $Q$ used by PLATE. As written, the theory supports the general design philosophy more than the concrete algorithm.
- The paper lacks key control ablations that are needed to empirically justify the proposed choices. In particular, it would be important to compare against matched-capacity variants with random $B$, random frozen orthogonal $Q$, and random $B$+$Q$.
- The neuron-level redundancy analysis may be too coarse. In particular, parameter-space similarity does not necessarily imply functional redundancy. For example, when it comes to the FFN layer in a Transformer, prior work has proposed a key-value memory view of FFNs: one projection can be interpreted as storing keys that detect input patterns, while the other projection provides values that determine the downstream contribution [1]. Under this view, neurons with similar key vectors can still be associated with different values and therefore play different functional roles. As a result, selecting neurons mainly based on weight-space colinearity does not by itself strongly justify the claim that these neurons are functionally redundant or especially safe carriers of plasticity. The current experiments suggest that the method may work in practice, but this explanatory story remains under-validated.

----
[1] Mor Geva, Roei Schuster, Jonathan Berant, and Omer Levy. *Transformer Feed-Forward Layers Are Key-Value Memories*. In Proceedings of the 2021 Conference on Empirical Methods in Natural Language Processing (EMNLP), pages 5484--5495, 2021.

---

> ### Author Rebuttal · Authors · 2026-03-30
>
> We thank the reviewer for his/her insightful comments drastically improving the understanding of the method. Results/Ablations/New figures will be added to the paper.
>
> **1. Matched-capacity random controls**
>
> We ran the requested matched-capacity controls on MNIST 0–4 / 5–9 and WikiText-2 / Middle English.
>
> Overall, the ablations suggest that the two components play different roles. The choice of $Q_{\text{in}}$ primarily affects retention: replacing PLATE’s geometry-aware subspace with a random orthogonal basis consistently increases forgetting on both domains, and randomizing both (B) and (Q) also degrades retention. The choice of $B$ mainly affects **adaptation efficiency**: random-(B) has little effect on forgetting, while least-redundant-(B) consistently hurts new-task learning and is also worse on language retention.
>
> On MNIST, the complementary **high-energy-(Q)** control produces the largest forgetting, supporting the claim that retention depends on the specific PLATE subspace rather than on an arbitrary (k)-dimensional constraint. We have no room to report the other results on the other datasets, but they will be added to the paper.
>
> #### MNIST 0–4 $\rightarrow$ 5–9
>
> | Method          |    T1 retention | $\Delta$ vs PLATE |      T2 acc |
> | --------------- | --------------: | ----------------: | ----------: |
> | **PLATE**       | **99.2 ± 0.4%** |                 — | 93.7 ± 0.5% |
> | Random-(B)      |     99.2 ± 0.2% |            +0.06% | 93.6 ± 0.4% |
> | Random-(Q)      |     97.2 ± 1.9% |            -1.93% | 95.0 ± 0.5% |
> | Random-(B+Q)    |     98.5 ± 0.5% |            -0.67% | 95.1 ± 0.6% |
> | Least-red-(B)   |     99.3 ± 0.2% |            +0.14% | 92.5 ± 0.9% |
> | High-energy-(Q) |     95.5 ± 2.9% |            -3.64% | 95.7 ± 0.7% |
>
>
> **2. Weight-space similarity vs. functional redundancy**
>
> We agree that weight-space similarity is only a heuristic proxy for functional redundancy, not a guarantee of functional interchangeability.
>
> This is clear in the continuous piecewise-affine view of MLPs:
> $\mathrm{MLP}(x)=\sum_{\omega\in\Omega}(A_{\omega}x+b_{\omega}),\mathbf{1}_{{x\in\omega}}.$
>
> Here, input-side weights determine the partition ($\Omega$), while output-side weights determine the affine map within each region. Two neurons can therefore have highly colinear input weights yet very different output weights, and hence very different functional roles.
>
> In the strictly data-free setting, grouping neurons by weight colinearity provides a practical way to identify degrees of freedom that are less likely to create new input partitions. Concentrating plasticity on these redundant directions is thus a geometry-based bias toward smaller functional drift, not a claim of exact functional equivalence. We will revise the paper to make this framing explicit.
>
> **3. Direct evidence that PLATE reduces old-model drift**
>
> We provide direct drift measurements on both tasks, comparing PLATE to matched-capacity internal controls with the same architecture but randomized design choices: rand-(Q) and rand-(B+Q)
>
> #### MNIST:  KL on old-distribution inputs
> we measure old-model drift as the average KL divergence between the old model’s output distribution and the adapted model’s output distribution on Task-1 test inputs.
>
> | Rank | (\tau) | PLATE | rand-(Q) | rand-(B+Q) |
> | ---: | -----: | ----: | -------: | ---------: |
> |    8 |   0.90 | 0.072 |    0.123 |      0.091 |
> |    8 |   0.95 | 0.026 |    0.106 |      0.122 |
> |    8 |   0.98 | 0.009 |    0.076 |      0.071 |
> |   16 |   0.90 | 0.096 |    0.150 |      0.097 |
> |   16 |   0.95 | 0.074 |    0.235 |      0.176 |
> |   16 |   0.98 | 0.014 |    0.136 |      0.111 |
> |   32 |   0.90 | 0.138 |    0.159 |     0.071 |
> |   32 |   0.95 | 0.186 |    0.252 |     0.131 |
> |   32 |   0.98 | 0.020 |    0.274 |      0.081 |
>
> PLATE achieves substantially lower predictive KL than rand-(Q) across all settings. At large rank, rand-(B+Q) can occasionally show low KL because the random projections suppress updates overall, rather than preserving useful old knowledge.
>
> WikiText-2 -> Middle English: old-distribution PPL degradation
>
> | Rank | PLATE | rand-(Q) | rand-(B+Q) |                           ME PPL  |
> | ---: | ----: | -------: | ---------: | ----------------------------------------------: |
> |   16 | +2.46 |    +3.12 |      +3.12 | PLATE 15.29 / rand-(Q) 15.34 / rand-(B+Q) 15.79 |
> |   32 | +4.28 |    +5.15 |      +4.57 | PLATE 10.82 / rand-(Q) 10.84 / rand-(B+Q) 11.13 |
> |   64 | +5.76 |    +6.51 |      +5.95 |    PLATE 7.66 / rand-(Q) 8.04 / rand-(B+Q) 8.13 |
>
> Across all tested ranks, PLATE shows lower old-distribution drift than both random controls while matching or improving new-task performance.

---

> > ### Author Rebuttal · Reviewer_qzU7 · 2026-04-03
> >
> > I appreciate the authors' detailed rebuttal. The added matched-capacity control ablations, including random $B$, random $Q$, random $(B+Q)$, as well as complementary controls such as least-redundant $B$ and high-energy $Q$, address the main empirical concern in my original review. I also appreciate the added direct drift measurements and the clearer statement that weight-space similarity is only a heuristic proxy for functional redundancy rather than a guarantee of functional equivalence.
> >
> > Overall, these additions substantially strengthen the empirical case for the method and make the paper's mechanism much better supported than in the original submission. In particular, the new results suggest that $Q$ is the main driver of retention, while $B$ appears to play a weaker and somewhat different role. I encourage the authors to make this asymmetry more explicit in the final version, but my main concerns have been adequately addressed.

---

> > > ### Author Response · Authors · 2026-04-03
> > >
> > > Thank you for noting that your concerns are fully resolved. Since the portal hasn't updated, we wanted to check if you still intended to adjust the score to help support the paper in the final decision.

---

### Official Review · Reviewer_p5Rc · 2026-03-13

**Soundness:** 3
**Presentation:** 2
**Significance:** 2
**Originality:** 3
**Overall Recommendation:** 4
**Confidence:** 2

**Summary:**

This paper offers a replay free (old data) and parameter efficient method PLATE for language model continual learning. Though the theoretical analysis of functional drift and loss curvature, PLATE strategically constraints weights updates using structural redundancy in pretrained networks to overcome forgetting. Empirical validations further support the effectiveness of PLATE over knowledge preserving and new task adaptation.

**Compliance With Llm Reviewing Policy:**

Affirmed.

**Final Justification:**

My concerns about empirical scope have been addressed so I changed my recommendation from weak reject to weak accept.

**Key Questions For Authors:**

I appreciate the author’s research, and I would be happy to adjust my rating if my concerns are addressed. Please see the weakness part.

**Limitations:**

yes.

**Strengths And Weaknesses:**

**Strength**:

The methodology of this paper is gounded in a rigorous geometric analysis that links the funcitonal drift and the restricted curvature of the original task loss. Such perspective explains the limitations of approximate orthogonality and motivating the need for low-drift update families. The authors provide empirical validation across both controlled in-distribution benchmarks and complex OOD scenarios for LLM, utilizing diverse performance metrics and local drift visualizations to demonstrate effectiveness.

------

**Weakness**:

1. More classic CL baselines should be used as baselines to strengthen the paper's basline. Including replay-free methods (regularization based) and other algorithms that utilize orthogonality.

2. More LLM CL baselines using PEFT modules should also be included for comparisons and more indepth discussion.

3. (Presentation)  Some texts in the figures (Fig.2, 3, 4, 5) are too small compared to Fig 1 and main texts, violating the manuscript formatting requirements of ICML, and affects reading.

---

> ### Author Rebuttal · Authors · 2026-03-30
>
> We thank the reviewer for helping us improve the quality of the paper. We acknowledge and changed the formatting issues. Regarding 1. and 2. we could run again some benchmarks on two additional methods, both replay-free, one being regularization-based and the other one orthogonality based: L2-init and O-LoRA. Please find below the results for each dataset.  All the plots for the experiments we ran have been updated in the paper, here is a quick summary to help you understand the results:
>
> **AG News→IMDB**
> This is a "free lunch" dataset: all adapter methods learn Task2 perfectly (100%) while retaining Task1 better than Full FT. L2 init actually hurts here: it caps learning at 91.3% and has the worst retention. The regularization is too rigid for large distribution shifts. LoRA and O-LoRA are essentially identical on this dataset.
>
> **MNIST 0-4 -> 5-9**
> L2 Init matches PLATE on retention (~98%) but requires all 100% of parameters, offering no parameter efficiency advantage. O-LoRA, despite using the same parameter budget as standard LoRA at every rank, consistently forgets more: retention degrades from 88% at r=8 down to 76% at r=128, converging toward full fine-tuning behavior.
>
> **Synthetic Regression**
> As distance increases, the methods diverge sharply. L2 init is the best at preventing forgetting across the entire range, reaching only 0.005 forgetting at maximum distance, an order of magnitude below Full FT (0.228), but at the cost of higher task 2 loss (0.0016 vs. 0.0003), reflecting the tension between regularization strength and plasticity. PLATE also controls forgetting very effectively (0.041 at max distance), far below Full FT and LoRA, while achieving better task 2 loss than L2 Init (0.0014 vs. 0.0016), suggesting it strikes a better plasticity–stability balance. LoRA and O-LoRA both fail to control forgetting.
>
>
> **WikiText-2 → Middle English**
> O-LoRA is the most notable failure: despite sharing the exact same parameter budget as standard LoRA at every rank, its forgetting is 10–100× worse, suggesting orthogonal initialization actively destabilizes original representations under large domain shifts. L2 init diverged across all λ values we tested, regardless of regularization strength, full fine-tuning on a 3B model either forgets catastrophically or fails to learn the new task, making it impractical in this setting.

---

> > ### Author Rebuttal · Reviewer_p5Rc · 2026-04-04
> >
> > Thanks for the rebuttal. My main concerns about the empirical scope have been well addressed, so I am increasing my score to 4. I have also read the other reviewers' comments and strongly suggest incorporating the additional results and clarifications into the revised manuscript.

---

### Decision · Program_Chairs · 2026-04-30

**Decision:**

Accept (regular)

**Comment:**

This paper proposes PLATE, a data-free and parameter-efficient continual adaptation method for pretrained models that uses geometric redundancy in pretrained weights to construct structured low-rank updates, with the goal of explicitly controlling the retention–plasticity trade-off without access to old-task data. The reviewers viewed the problem as important and practically relevant, and generally found the combination of geometric motivation, structured adapter design, and empirical results promising. The review set was overall supportive, with one reviewer explicitly at Accept and others moving to or remaining at Weak Accept after rebuttal. The main concerns were the limited baseline coverage in the original submission, the gap between the theoretical motivation and the concrete geometry-aware construction, the need for stronger matched-capacity control ablations and more direct drift evidence, and questions about longer-horizon continual behavior, activation-vs-weight geometry, and robustness under model compression. In the rebuttal, the authors addressed these concerns substantively by adding replay-free and orthogonality-based baselines, providing matched-capacity random-control ablations for B and Q, reporting direct old-model drift measurements, clarifying that weight-space similarity is only a heuristic proxy for functional redundancy, describing the recursive protocol for longer task sequences, showing multi-task behavior beyond two tasks, and adding evidence that the method degrades gracefully under compression rather than failing abruptly; these additions were received positively, and multiple reviewers indicated that their main concerns were fully resolved and increased their scores. Overall, I find that the paper makes a meaningful and well-supported contribution to data-free continual adaptation for pretrained models, and I recommend Accept.